# Granulosa cell transcription is similarly impacted by superovulation and aging and predicts early embryonic trajectories

Klaudija Daugelaite [1,2,7], Perrine Lacour [2,3,7], Ivana Winkler [3,7], Marie-Luise Koch[1], Anja Schneider [1], Nina Schneider[3], Francesca Coraggio[3], Alexander Tolkachov[1,6], Xuan Phuoc Nguyen[4], Adriana Vilkaite [4], Julia Rehnitz [4], Duncan T. Odom [1] ✉ & Angela Goncalves [3,5] ✉

In vitro fertilization efficiency is limited in part because a fraction of retrieved oocytes fails to fertilize. Accurately evaluating their quality could significantly improve in vitro fertilization efficiency, which would require better understanding how their maturation may be disrupted. Here, we quantitatively investigate the interplay between superovulation and aging in mouse oocytes and their paired granulosa cells using a newly adapted experimental methodology. We test the hypothesis that superovulation disrupts oocyte maturation, revealing the key intercellular communication pathways dysregulated at the transcriptional level by forced hormonal stimulation. We further demonstrate that granulosa cell transcriptional markers can prospectively predict an associated oocyte's early developmental potential. By using naturally ovulated old mice as a non-stimulated reference, we show that aging and superovulation dysregulate similar genes and interact with each other. By comparing mice and human transcriptional responses of granulosa cells, we find that age-related dysregulation of hormonal responses and cell cycle pathways are shared, though substantial divergence exists in other pathways.

Mammalian reproduction depends on the production and fertilization of high-quality oocytes. During mammalian ovulation, oocytes are released into the female reproductive tract as cumulus-oocyte-complexes (COCs), each consisting of an oocyte (OC) surrounded by somatic granulosa cells (GCs). In humans, reproduction is naturally inefficient with high rates of aneuploidy in oocytes and frequent loss of early pregnancy[1,2]. Common pathologies such as ovulatory disorders, endometriosis, occlusions of the fallopian tubes, uterine fibroids, or sperm anomalies further decrease fertility[3]. To treat infertility, assisted reproductive technologies such as ovarian stimulation (or superovulation) followed by in vitro fertilization (IVF) have been developed[4,5].

Human IVF typically involves ovarian stimulation using hormonal treatments, usually with gonadotropins, to induce the development of multiple oocytes. While this increases the number of oocytes available for retrieval, there is some concern that the rapid stimulation and altered hormonal environment might affect the quality of some oocytes[6–8]. However, disentangling the effect of numerous individual factors (e.g., patient age, lifestyle, medical history, genetic background, and/or sperm quality) from the process of superovulation

[1]Division of Regulatory Genomics and Cancer Evolution, German Cancer Research Center (DKFZ), Heidelberg, Germany. [2]Faculty of Biosciences, Ruprecht-Karl-University Heidelberg, Heidelberg, Germany. [3]Division of Molecular and Computational Prevention, German Cancer Research Center (DKFZ), Heidelberg, Germany. [4]Department of Gynecological Endocrinology and Fertility Disorders, University Women's Hospital Heidelberg, Heidelberg, Germany. [5]Medical Faculty Mannheim, Heidelberg University, Heidelberg, Germany. [6]Present address: Department of Cardiology, Angiology and Pneumology, Internal Medicine III, Heidelberg University Hospital, Heidelberg, Germany. [7]These authors contributed equally: Klaudija Daugelaite, Perrine Lacour, Ivana Winkler. ✉ e-mail: d.odom@dkfz.de; a.goncalves@dkfz.de

itself is very challenging in humans. Model organisms like rodents or livestock can be used as a better controlled system to study how oocytes respond to superovulation. In these organisms, studies based on morphological and biochemical characterization have confirmed that, when compared to natural ovulation, a fraction of the oocytes released after superovulation is more immature[9,10], shows impaired epigenetics[11,12], and lower fertilization and implantation rates[13,14].

Follicle maturation is a critical and complex process that prepares the egg for successful fertilization and subsequent embryonic development. Initially, oocytes in the ovary are arrested in the prophase stage of the first meiotic division. As follicles mature, the oocyte grows and stockpiles mRNA, before becoming transcriptionally silent around the antral and preovulatory stages[15,16]. A surge of luteinizing hormone (LH) triggers both ovulation and the resumption of meiosis in the oocyte. The oocyte completes the first meiotic division and begins the second meiotic division, but halts at metaphase II, where it remains arrested until fertilization[17]. In parallel to meiotic resumption, widespread re-poly-adenylation and degradation of transcripts[18] shift the expression profile of the still transcriptionally-inactive cell to prepare for early embryonic development.

Proper oocyte growth and maturation requires correct bi-directional communication with the surrounding cumulus granulosa cells[19–21]. It has thus been speculated that an oocyte and derived embryo's quality could be evaluated non-invasively by the transcriptional patterns in the surrounding granulosa cells (reviewed in Li et al.[22]). In human patients, clinical studies have collected and tracked the granulosa cells that are usually discarded in IVF protocols. Identifying a reliable transcriptomic signature connecting granulosa cells to clinical outcome has however remained elusive due to variation in IVF procedures and analysis, pooling of COCs, small sample numbers, and lack of replicability between studies (reviewed in Uyar et al.[23] and by Scarica[24]). Analogous studies have been conducted in cattle[25], but similarly to humans, no consensus markers in granulosa cells have been successfully identified that predict oocyte quality. To our knowledge, such an approach has not been performed in mice.

One of the main factors leading to decreased oocyte quality is aging. The recent increase in average reproductive age in many countries, coupled with natural age-related decline in fertility[26,27], has led clinicians to extend the use of IVF technologies to increasingly older patients. In human, fertility sharply decreases with age due to reduced oocyte numbers[28] and poorer oocyte quality, reflecting gene expression dysregulation and/or inadequate maturation[29,30]. Similarly, in mice, oocytes from old individuals have increased aneuploidy rates[31,32], gene expression dysregulation[31,33] and lower developmental potential[34].

How aging shapes the response to superovulation also remains poorly understood, often due to model limitations. COC aging studies commonly rely on superovulation to obtain old samples without natural ovulation controls. It is thus impossible to study the impact of aging alone or how it impacts superovulation response, as both would require naturally ovulated COCs from old individuals. To our knowledge, there are no prior studies that decouple the effects of superovulation and natural aging in ovulated cumulus-oocyte complexes.

Here, we compare and quantify the effects of aging and super-ovulation on ovulated COCs, and demonstrate that oocyte early developmental potential can be predicted by profiling surrounding granulosa cells. To accomplish this, we employed a rigorous study design by isolating individual granulosa-oocyte pairs and including naturally ovulated COCs from young and old mice. Our improved study design identified the perturbations in oocyte maturation and cell-cell communication driven by hormonal stimulation and aging, discovered granulosa cell markers that can predict the early developmental potential of oocytes, and showed that aging and super-ovulation lead to similar transcriptomic perturbations.

## Results

### Dissociating cumulus-oocyte complexes into paired components

In order to simultaneously analyze oocyte and granulosa cell transcriptomes from individual COCs, we adapted isolation protocols to manually separate the two while retaining their in vivo pairing information ("Methods", Fig. 1a, Supplementary Fig. 1a, Supplementary Movie 1). Using a SMART-seq2 protocol, each oocyte was sequenced as a single cell and its surrounding granulosa cells as a small bulk ("Methods"). For each experimental condition and cell type, we collected 15–31 replicates, and in 72% of these replicates both paired components successfully passed quality control (Supplementary Data files 1 and 2). To increase statistical power, non-paired oocytes and/or granulosa cells were included in specific analyses, as described below.

We collected paired COC transcriptomes from 3-month old (henceforth "young") and 12-month old (henceforth "old") mice in four conditions: naturally-ovulated young (NY), superovulated young (SY), naturally-ovulated old (NO), and superovulated old (SO) (Fig. 1b). The oocytes and granulosa cells from all conditions were readily distinguished by their full transcriptomes (Fig. 1c, Supplementary Fig. 1b, c). We confirmed no cross-contamination between oocytes and granulosa cells by inspecting the transcription of previously known marker genes[35–38] (Fig. 1d). Exploiting the high number of replicates, we identified multiple oocyte- and granulosa-specific markers with similarly polarized gene expression (Fig. 1d, Supplementary Data 3).

### Superovulation perturbs key COC maturation pathways

We reasoned that any developmental immaturity from superovulation should be reflected in simultaneous dysregulation of related pathways in oocytes and their associated granulosa cells. In young mice, we first compared the impact of superovulation on all available oocyte and granulosa cell transcriptomes. In oocytes, we identified 3519 genes (of 22,181 genes detected) that were differentially expressed following superovulation (adjusted $p < 0.05$, 46% upregulated) (Fig. 2a). As expected, many pathways critical for oocyte development were perturbed (Fig. 2a, Supplementary Data 4), including meiosis, mitochondrial metabolism, DNA damage, and hormonal response. In granulosa cells, superovulation dysregulated 2765 genes (of 25,201 detected) (adjusted $p < 0.05$, 40% upregulated) (Fig. 2b). Similarly to oocytes, cell-cycle, hormonal and damage response pathways were enriched in differentially expressed genes (Fig. 2b, Supplementary Data 4). Overall, superovulation transcriptionally disrupts pathways required for successful COC development.

In addition to the unbiased analysis above, we compared the expression of genes involved in three mechanisms known to regulate oocyte function (cytoplasmic maturation, spindle assembly, and cell-cell communication) between natural and superovulated oocytes and their granulosa cells.

First, we inspected cytoplasmic maturation, and asked whether superovulation perturbs the expected re-poly-adenylation, de-adenylation, and degradation of known transcripts[39–43]. To do so, we compared a poly-A-biased protocol (SMART-seq2) and a total RNA-seq protocol based on random priming of transcripts ("Methods", Supplementary Data 5). An impairment of re-poly-adenylation of transcripts from superovulation would result in an apparent downregulation of these genes in the poly-adenylated biased experiments, whereas no difference would be observed between mature and immature oocytes in total transcriptome experiments (Fig. 2c, re-poly-adenylation). Superovulated disruption of de-adenylation would create a symmetric pattern (Fig. 2c, de-adenylation). In contrast, degradation of transcripts should result in the same trend with both protocols (Fig. 2c, degradation).

Transcripts were selected based on previous studies on these processes during oocyte maturation. A recent transcriptome-wide study quantified poly-A tail length and transcript level during oocyte

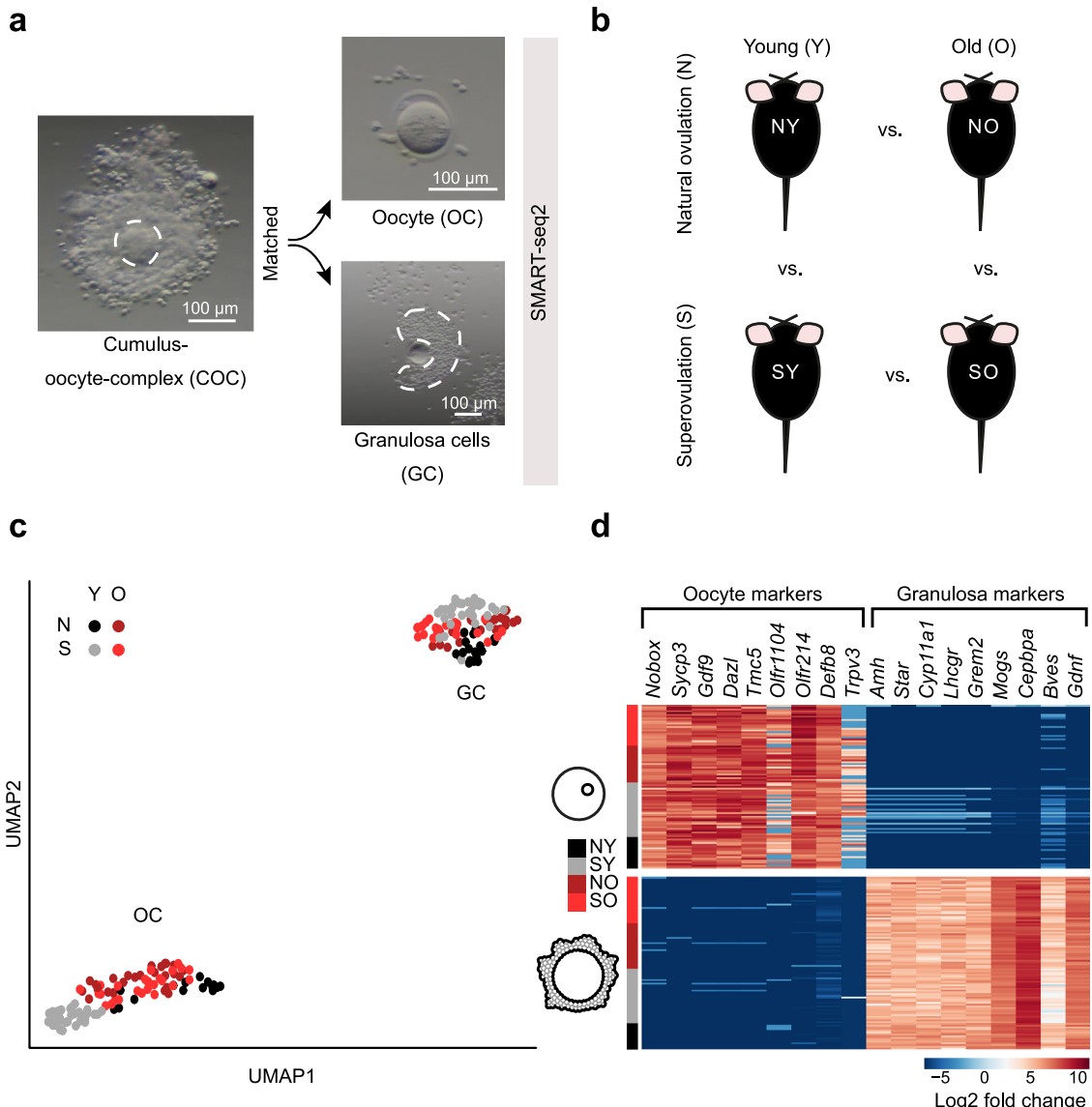

**Fig. 1 | Trackably cleaving cumulus-oocyte complexes into their two component cell types in naturally and superovulated young and old mice.**
**a** Representation of manual separation of cumulus-oocyte complexes (COCs) into oocyte (OC) and small bulk of its granulosa cell (GC) pairs, followed by single cell sequencing using SMART-seq2 ("Methods"). Experiments were repeated independently for each mouse ($n = 13$), in total 98 times (SMART-seq2 experiments). **b** COCs were collected from naturally ovulated young (NY, $n = 3$), superovulated young (SY, $n = 4$), naturally ovulated old (NO, $n = 3$), and superovulated old (SO, $n = 3$) mice. **c** Principal component analysis (PCA) representation of all OC ($n = 98$) and GC ($n = 93$) samples from all four experimental groups (NY−black, SY−gray, NO−dark red, SO−light red). **d** Expression of cell-type-specific markers in OCs and GCs in all experimental groups (NY−black, SY−gray, NO−dark red, SO−light red). First four markers for each cell type were selected from literature, and the last five were identified from our data. Source data are provided as a Source Data file.

maturation[43], while others have targeted specific processes or genes[39–41]. Using the transcriptome-wide reference[43], we show that these processes are widely disrupted in superovulated oocytes (Supplementary Fig. 1d). Disrupted genes include post-transcriptional regulators *Dazl*, *Cnot7*, and *Btg4* (Fig. 2d, Supplementary Fig. 1e), which were previously identified as targets of re-poly-adenylation during oocyte maturation[18,39,41,42,44]. Similarly, we found that superovulation disrupts the de-adenylation of *Smc4*, a chromosome condensation gene whose poly-A tail is usually removed during oocyte maturation[40], and of *Taok2* and *Isyna1*, identified recently as de-adenylated but not degraded[43]. Finally, we found that the degradation of transcripts during oocyte maturation is also disrupted by superovulation, including the 3′-UTR binding protein *Cpeb1*[39,42]. These results demonstrate that superovulation disrupts multiple interlinked mechanisms of transcriptome remodeling active in oocyte maturation.

Second, we asked whether superovulation perturbs the regulation of spindle assembly checkpoint (SAC) genes involved in meiosis. Genes involved in chromosome attachment to microtubules (*Mad2l1*, *Mad2l2*) were downregulated, while those responsible for accurate chromosome segregation (*Bub1*, *Bub1b*, and *Ttk*) were upregulated, which suggests differential SAC deployment in superovulated oocytes (Fig. 2e).

Third, we evaluated whether superovulation perturbs the cell-cell communication from granulosa cells controlling meiotic resumption in oocytes. In normal conditions, the LH surge transcriptionally upregulates *Areg* and downregulates *Esr2* and *Npr2* in granulosa cells. Subsequently, there is a decrease of NPR2 receptor activity and related increase in PDE5 enzymatic activity, which in turn leads to degradation of cGMP and the resumption of meiosis[45]. In our data, superovulation impaired the transcriptomic response of

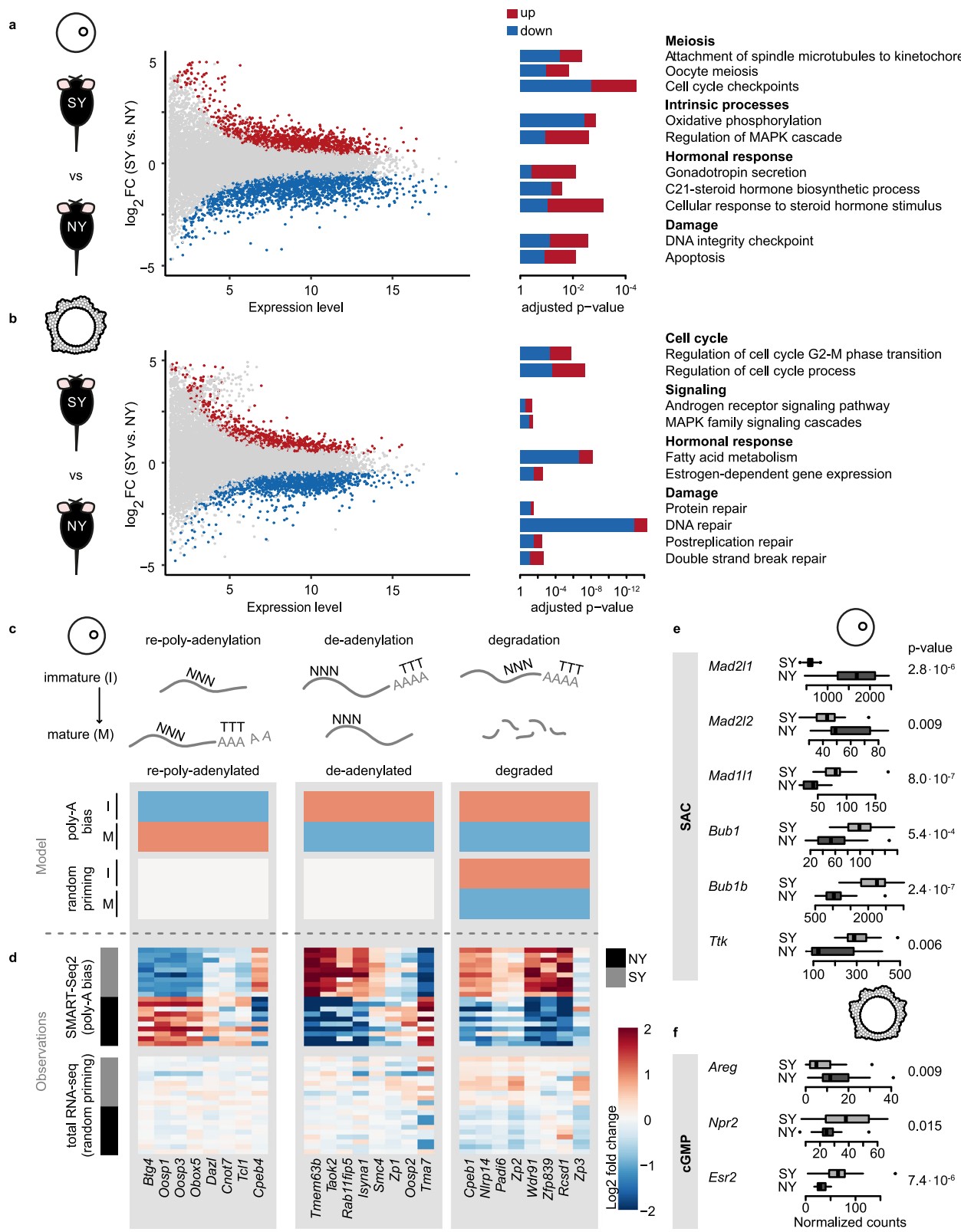

granulosa cells, potentially delaying the transmission of the maturation cues (Fig. 2f). Because it is highly expressed, we were able to validate that *Esr2* expression is increased in superovulated granulosa cells using Hybridization Chain Reaction (HCR) as an orthogonal imaging-based methodology ("Methods", Supplementary Fig. 1f). Attempts to validate *Areg* and *Npr2* using HCR were unsuccessful due to their lower expression.

In conclusion, superovulation perturbs germ and somatic cell transcription of key genes involved in COC maturation pathways.

### Grouping of superovulated COCs by granulosa transcription

We reasoned that the perturbations caused by superovulation would be first reflected within the extensive granulosa cell-oocyte communications which regulate oocyte maturation. We specifically designed

**Fig. 2 | Superovulation disrupts the expression of genes involved in oocyte maturation.** Differentially expressed genes and enriched functional pathways between naturally ovulated (NY) and superovulated young (SY) oocytes (OCs) (**a**) and granulosa cells (GCs) (**b**) (*n* = 3 NY, 4 SY mice). The bars are colored based on the proportion of genes in the pathways that are up- or downregulated in SY compared to NY. Differential gene expression was performed using a two-tailed Wald test with multiple testing correction as implemented in DESeq2. Over-representation of pathways was done using a one-tailed hypergeometric test and *p* values were corrected for multiple testing. **c** Schematic representation of gene expression expectation in mature (M) and immature (I) oocytes between poly-A biased (SMART-seq2) and unbiased protocols (total RNA-seq) for genes involved in oocyte transcriptional remodeling during maturation (re-poly-adenylation, de-adenylation, and degradation). Expectation of upregulation is represented in light red, downregulation in light blue. **d** Comparison of gene expression between naturally and superovulated oocytes using SMART-seq2 (*n* = 15 NY, 31 SY) and total RNA-seq (*n* = 25 NY, 22 SY). For each technology, the fold change is computed between the two groups ("Methods"). Each row represents gene expression in an individual oocyte (randomly subsampled set). Full heatmap with additional genes available in Supplementary Fig. 1e. **e** Expression of genes involved in spindle assembly checkpoint (SAC) machinery in naturally and superovulated oocytes (*n* = 15 NY, black, 31 SY, gray, oocytes). *P* values were computed using a two-tailed Wilcoxon test. Center line—median, box limits—first and third quartiles, whiskers—maximum and minimum or 1.5 times interquartile range if outliers, dots—outliers. **f** Expression of genes involved in the cGMP pathway in naturally and superovulated granulosa cells (*n* = 18 NY, 31 SY granulosa cells). *P* values were computed using a two-tailed Wilcoxon test. Center line—median, box limits—first and third quartiles, whiskers—maximum and minimum or 1.5 times interquartile range if outliers, dots—outliers. Source data are provided as a Source Data file.

our experiments with this hypothesis in mind and kept track of the in vivo pairing of oocytes and granulosa cells. For each young COC, we constructed a cell-cell communication profile based on co-expression of ligand-receptor pairs ("Methods"). Naturally ovulated COCs grouped together based on hierarchical clustering, indicating homogenous cell-cell communication (Fig. 3a, Supplementary Fig. 2a). In contrast, we observed that superovulated COCs split into two distinct clusters (clusters $S_N$ and S), where $S_N$ more closely resembles naturally ovulated COCs. We further confirmed the relative similarity of $S_N$ and N by dimensionality reduction analysis (Fig. 3b).

Next, we reasoned that the similarity of $S_N$ and normal COCs should be reflected in their corresponding transcription factor (TF) activities. First, we identified the transcription factors upstream of dysregulated COC pathways after superovulation, and calculated each TF's activity score using the expression level of its target genes ("Methods", Fig. 3c, d). For some TFs, such as *Esr1* and *Esr2*, the number of expressed direct target genes was too low to accurately assess their activity and the TFs were thus excluded. In oocytes, TF activity was grouped into natural and superovulated clusters without an obvious substructure (Supplementary Fig. 2b), as expected due to lack of active oocyte transcription at this meiotic stage[46].

In contrast, superovulated granulosa cells again formed two distinct subclusters (clusters $S_N$ and S, Fig. 3d), paralleling the cell-cell communication results. These clusters were reproducible after bootstrapping and subsampling, and were not the result of inter-individual variability (Supplementary Fig. 2c). An orthogonal pathway activity analysis further confirmed that $S_N$ granulosa cells were highly similar to normally ovulated cells in postreplication and protein repair, MAPK and estrogen signaling, and had intermediate activity in cell cycle control, DNA repair and androgen receptor signaling (Fig. 3e, Supplementary Fig. 2d). Related pathways were also overrepresented in differentially expressed genes between the two groups (Supplementary Fig. 2e, Supplementary Data 6). Fatty acid metabolism and double strand break repair pathways were more similar between $S_N$ and S compared to N. In all further analyses, we defined consensus clusters for N, $S_N$, and S by including the 34 out of 40 granulosa cell samples where cell-cell communication and TF activity analyses were in agreement.

**Granulosa markers predict oocyte developmental trajectory**
We hypothesized that the two granulosa clusters may be linked to different embryo development outcomes. To link granulosa cell transcriptional state to early embryo development, we developed a targeted IVF assay (Fig. 4a, "Methods"). Briefly, we fertilized and cultured single oocytes until early morula and late blastocyst stages, while in parallel sequencing their associated granulosa cells. This large, independent dataset arose from 219 COCs from 9 naturally and 8 SY mice (Supplementary Data 7). We found a 91.5% average fertilization rate for naturally ovulated oocytes and 87.9% for superovulated oocytes. We then assayed the transcriptomes of the resulting embryos (*n* = 154, Supplementary Fig. 3a).

We trained a support-vector machine (SVM) classifier using the original 34 granulosa cell replicates (Figs. 3, 83.3% accuracy on test set, 5/6 embryos correctly classified, Supplementary Fig. 3b), which we then used to assign granulosa cells from these new IVF experiments into $S_N$, S and "not assignable" (NA) (Fig. 4a, "Methods"). We transferred the labels from the classified granulosa cells to their corresponding 154 embryos, and then assessed the embryo developmental trajectory, including arrested state and copy number variation (CNV).

Of the 103 superovulated embryos, 14 stopped developing before reaching the 8-cell stage (Supplementary Fig. 3a), 11 of which were classified as S or "not assignable" (Supplementary Fig. 3c). Due to the small number of arrested embryos, this was not statistically significant. Only three embryos had large enough aneuploidy to be detected by gene expression data (Supplementary Fig. 3d); these showed no clear relationship with the classification assignment.

To more closely investigate the developmental timing of these embryos, we combined a previously published dataset of early embryonic development[47] with our morulas and blastocysts derived from natural ovulation (Fig. 4b). We then selected genes that closely correlated with developmental stage (Fig. 4c). Using these genes, we then ordered all embryos into a developmental trajectory ("Methods"). Embryos derived from natural ovulation were the most advanced (Fig. 4d), as previously described[48]. The embryos generated using superovulated oocytes clearly separated, with the $S_N$ embryos being more advanced and closer to the natural group (Fig. 4d, Supplementary Fig. 3e).

Importantly, the $S_N$ and S classes were readily discernible without the need to perform complex transcriptome-wide analyses, by using qPCR on genes from our SVM classifier, including *Dync1h1*, *Hook3*, *Kmt2a*, *Lrp6*, *Phf20l*, *Vcan*, and *Zfp451* (Fig. 4e, Supplementary Fig. 3f). These results demonstrate that transcriptional signatures of granulosa cells can be used to accurately predict the early developmental trajectory of their associated oocyte.

In sum, our method can reliably and non-invasively identify which superovulation-derived pre-implantation embryos most closely resemble those generated from naturally released oocytes. Thus, the original transcriptional state of a cumulus–oocyte complex is reflected in the gene expression of derived embryos, despite the massive genome remodeling occurring during fertilization and zygotic genome activation (reviewed in Jukam et al.[49] and Schulz and Harrison[50]).

**Aging and superovulation induce similar expression changes**
Since both superovulation and aging lead to lower quality oocytes and poor developmental outcomes, we compared the mechanisms involved. By introducing naturally ovulated cells from aged individuals, our experimental design newly permitted a rigorous comparison of how COC transcription changes during natural aging (that is, naturally ovulated young versus old cells) and upon superovulation (that is, naturally ovulated young versus superovulated cells) (Fig. 5a).

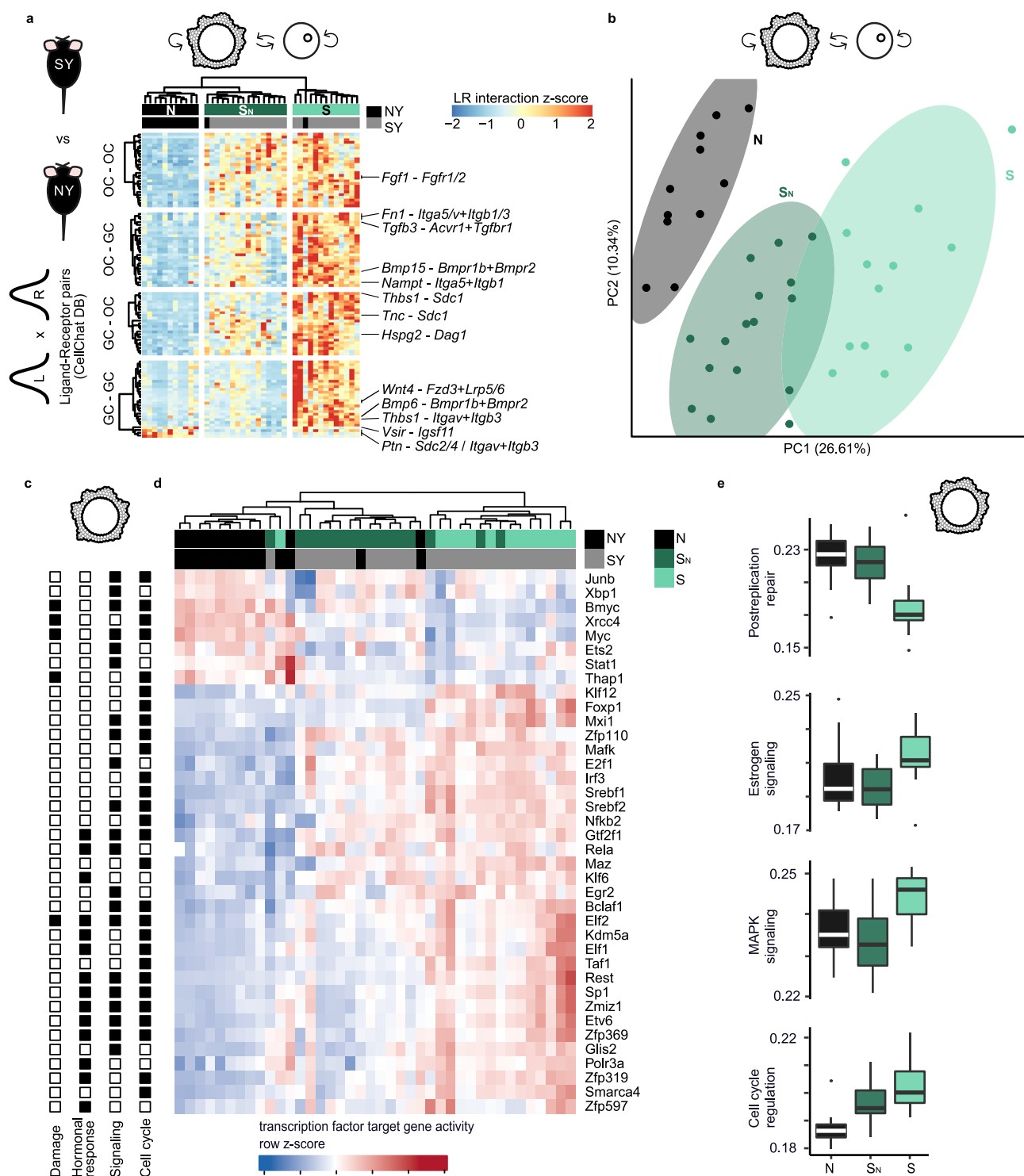

**Fig. 3 | Superovulation dysregulates ligand-receptor expression and transcription factor activity in cumulus-oocyte complexes. a** Cell-cell communication scores between oocyte (OC) and granulosa cells (GC) calculated as a product of ligand (L) and receptor (R) gene expressions ("Methods"). Interaction scores are shown for ligand-receptor pairs (rows, z-scaled) in young natural and superovulated cells (columns, $n = 13$ NY, black, 27 SY, gray, OC-GC pairs) that displayed significant change between the three clusters N (black), $S_N$ (dark green), and S (light green) (Kruskal-Wallis test, adjusted $p < 0.05$). Column clustering dendrogram shown was obtained from the full list of ligand-receptor pairs shown in Supplementary Fig. 2a. All non-annotated ligand-receptor pairs are related to extracellular matrix. **b** PCA computed on OC-GC interaction scores. **c** Transcription factors associated with the pathways shown to be enriched in superovulated GCs in Fig. 2b.

Significant association with pathway categories (DNA damage, hormonal response, signaling, or cell cycle) is indicated as a filled square (one-tailed hypergeometric test, adjusted $p < 0.05$). **d** Activity scores of transcription factors whose activity scores were significantly different between naturally and superovulated young GCs are shown (two-tailed permutation test, adjusted $p < 0.05$, "Methods", $n = 13$ NY, 27 SY GCs). **e** Activity scores of selected pathways enriched in superovulated GCs from Fig. 2b split by consensus clusters N, $S_N$, and S ($n = 10$ N, 12 $S_N$, 12 S). Activity scores for other pathways from Fig. 2b are shown in Supplementary Fig. 2d. Center line—median, box limits—first and third quartiles, whiskers—maximum and minimum or 1.5 times interquartile range if outliers, dots—outliers. Source data are provided as a Source Data file.

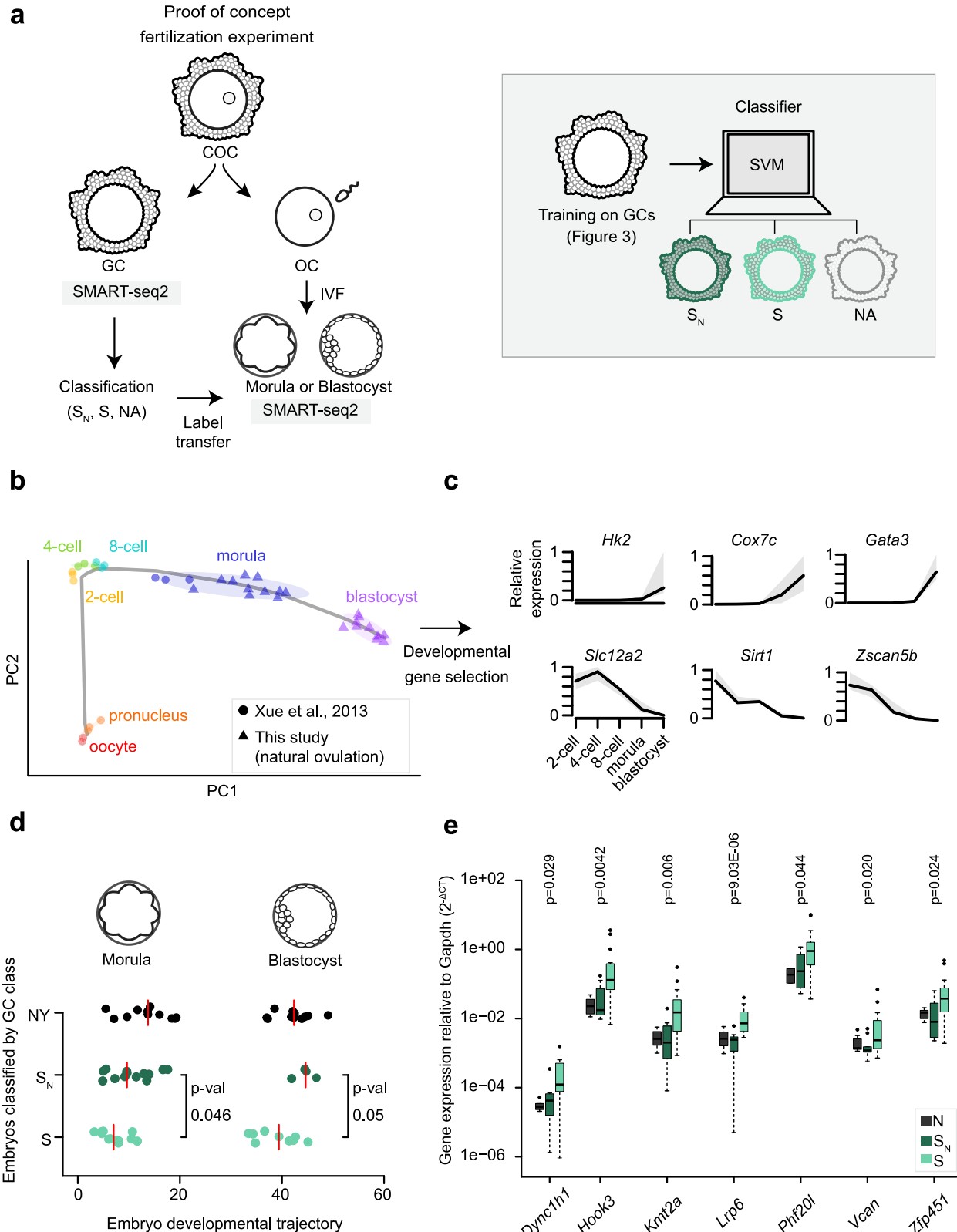

We identified 1410 genes (of 22,181 genes detected, 41% upregulated) that were differentially expressed (adjusted $p < 0.05$) during natural aging in oocytes, and 1816 (of 25,201 detected, 33% upregulated) in granulosa cells. We discovered that in oocytes and granulosa cells both superovulation and aging perturb a common set of genes (1008 in oocytes and 1132 in granulosa cells) and with a similar directionality (Figs. 5b, c, Supplementary Fig. 4a). Some of these common genes

were also used in the previously described classifier (Supplementary Fig. 4b).

Overrepresentation analysis of the differentially expressed genes shared between aging and superovulation was highly similar to what we found for superovulation alone. We observed an enrichment for pathways related to estrogen response, meiosis, oxidative phosphorylation, and response to oxidative stress (Supplementary Data 4). The

**Fig. 4 | Pre-implantation embryo quality prediction from granulosa cells.**
**a** Schematic representation of targeted in vitro fertilization (IVF) experiment and embryo classification by support vector machine (SVM) based on their granulosa cell (GC) transcriptomes ($S_N$–dark green, S–light green, not assigned (NA)–transparent, "Methods"). **b** Pre-implantation development reference samples from Xue et al.[47] (dots) and this study (triangles). **c** Examples of relative expression of selected developmental genes used in the pseudotime analysis (normalized to 9th decile). Center line–median, gray area limits–first and ninth deciles. **d** Embryos classified by their granulosa cell class along pseudotime calculated on

developmental genes ($n = 22$ natural young (NY), 17 $S_N$, 21 S embryos). Red line–median, two-tailed Wilcoxon test was used to compute $p$ values. **e** Gene expression of significant classifier genes in NY, $S_N$, and S granulosa cells measured by qPCR ($n = 7$ NY, 11 $S_N$, and 23 S granulosa cells). All tested genes are shown in Supplementary Fig. 3f. Two-tailed Wilcoxon test was performed between $S_N$ and S groups to determine $p$ values. Center line–median, box limits–first and third quartiles, whiskers–maximum and minimum or 1.5 times interquartile range if outliers, dots–outliers. Source data are provided as a Source Data file.

role of mitochondria and oxidative stress in oocyte aging are known, and mitophagy disruption has recently been highlighted as a likely critical factor for oocyte quality decrease in aging[51]. When specifically looking at mitophagy pathway genes, we observed similar perturbations in response to superovulation and aging (Supplementary Fig. 4c).

We asked whether aging may also increase cell-to-cell variability in oocytes, as has been previously reported for other cell types[52,53]. To do so, we first determined the gene-by-gene variability in gene expression using Shannon entropy, which is a measure of variability with a value between 0 (no variability) and 1 (high variability). We then calculated a differential Shannon entropy (difference between the Shannon entropy in one condition minus the other condition), thus measuring whether gene expression is more variable in one condition compared to the other ("Methods"). When comparing young and old oocytes (Fig. 5d), we found that aging increases cell-to-cell variability for many genes. Similar analyses revealed that superovulation also substantially increases oocyte cell-to-cell variability in gene expression (Fig. 5d). This was further validated using our total RNA-seq samples in oocytes after natural or superovulation (Supplementary Fig. 4d). In granulosa cells, aging also increases gene expression variability between samples, which is not observable in superovulation (Supplementary Fig. 4e).

To confirm that gene expression differences are reflected in upstream and downstream communication and TF usage, we then computed cell-cell communication and TF activity profiles for old COCs. We observed that old COCs locate between young natural and superovulated COCs in communication profile space (Fig. 5e). Likewise, TF activity profiles of old granulosa cells localized close to cluster $S_N$ in TF activity space (Fig. 5f).

Together, these results strongly suggest that superovulation and aging lead to qualitatively similar responses, targeting the same underlying regulatory networks to different degrees.

### Modest conservation of age-related changes in human samples
We asked whether similar dysregulation occurred in human granulosa cells and oocytes upon aging. Naturally ovulated human samples are unavailable. Therefore, we collected human granulosa cells from patients undergoing IVF, sorted by age into younger (<31 years old) versus older (>38 yo) groups, and performed bulk RNA-sequencing ("Methods"). To this dataset, we added publicly available oocyte gene expression data for younger (<27 yo) and older patients (>40 yo)[54].

We identified that dysregulation of a diversity of signaling pathways, including estrogen-dependent gene expression, was shared in mouse and human in granulosa cells, suggesting aging might similarly impact COC quality ("Methods", Fig. 5g). This analysis was robust to the exclusion of genes that are differentially expressed between human mural and cumulus cells ("Methods"), to correct for the fact that only cumulus cells were used in mice compared to mural granulosa cells in humans. The presence of different granulosa populations in human samples was because of differing cell collection protocols, where human granulosa cells were collected from ovarian punctures. Interspecies differences we observed included dysregulation of cell cycle and fatty acid metabolism divergence, among others ("Methods", Fig. 5g, Supplementary Data 8). In oocytes, a similar analysis comparing our aging results in mice to a previously published human dataset[54]

shows that pathways involved in mitochondrial metabolism, oxidative stress response, and different fertility processes respond similarly in both species to aging (Supplementary Fig. 4f).

### Superovulation and aging effects interact
Finally, we investigated whether the effects of superovulation and aging on mouse COCs are additive or whether there is a statistical interaction between the two (Fig. 5h). Surprisingly, among old oocytes and granulosa cells, we found very few significant differences in gene expression levels between naturally and superovulated cells (Supplementary Figs. 4g, h). This analysis had sufficient statistical power to identify any gene expression differences, as the number of replicates was comparable to the number of young mouse experiments (Supplementary Data 1 and 7). We observed that the impact of superovulation is reduced in old COCs, by projecting the transcriptomic profile of old cells onto a PCA computed using superovulation responsive genes in young cells (Fig. 5i, Supplementary Fig. 4k). Thus, aging and superovulation do not have an additive effect on gene expression. Indeed, we found that old COCs often exhibit gene expression profiles intermediate between young natural and young superovulated COCs, including maternal remodeling targets, cGMP cascade genes and SAC genes (Supplementary Fig. 4i, j).

To test the relative impact of age and superovulation, we quantified the shifts relative to PC1 between all four groups (SO, NO, SY, NY) (Fig. 5j, k, Supplementary Fig. 4l, m). Superovulation induces a strong transcriptional shift in young COCs, which is greatly attenuated in the old cells. Moreover, age-induced transcriptional changes differ between natural and superovulated COCs.

In sum, experiments performed with superovulated cells do not accurately capture the impact of aging that occurs in naturally ovulated COCs. This result could only be established by the incorporation of naturally ovulated old COCs to determine the interaction component.

## Discussion
Our study quantitatively dissects the impact of superovulation and aging on COCs, using mouse as a tractable mammalian model system. To analyze the functional interactions between oocytes and their surrounding granulosa cells, we utilized an experimental approach where the oocyte-granulosa pairs are separated while maintaining their pairing information. Our experimental design further incorporates COCs from naturally ovulated older mice, introducing a key control missing from previous studies. Our analyses reveal that superovulation (i) perturbs gene expression in maturation pathways, like hormonal response in granulosa cells, and meiosis and transcriptome remodeling in oocytes, (ii) produces two distinct classes of different quality young COCs, and (iii) qualitatively mimics the impact of aging.

Our transcriptomic approach confirmed that superovulation perturbs meiosis genes in oocytes[9,55–58]. Even though transcription is silent between the GVBD and MII stages, translation of many maternal mRNAs is upregulated at the GVBD-prometaphase transition and, at MII stage, large-scale translation takes place[39,40]. This includes SAC genes *Bub1b* and *Mad2l1*, and numerous cell cycle genes, while *Mad1l1* translation decreases[40,59,60], meaning that differences in expression

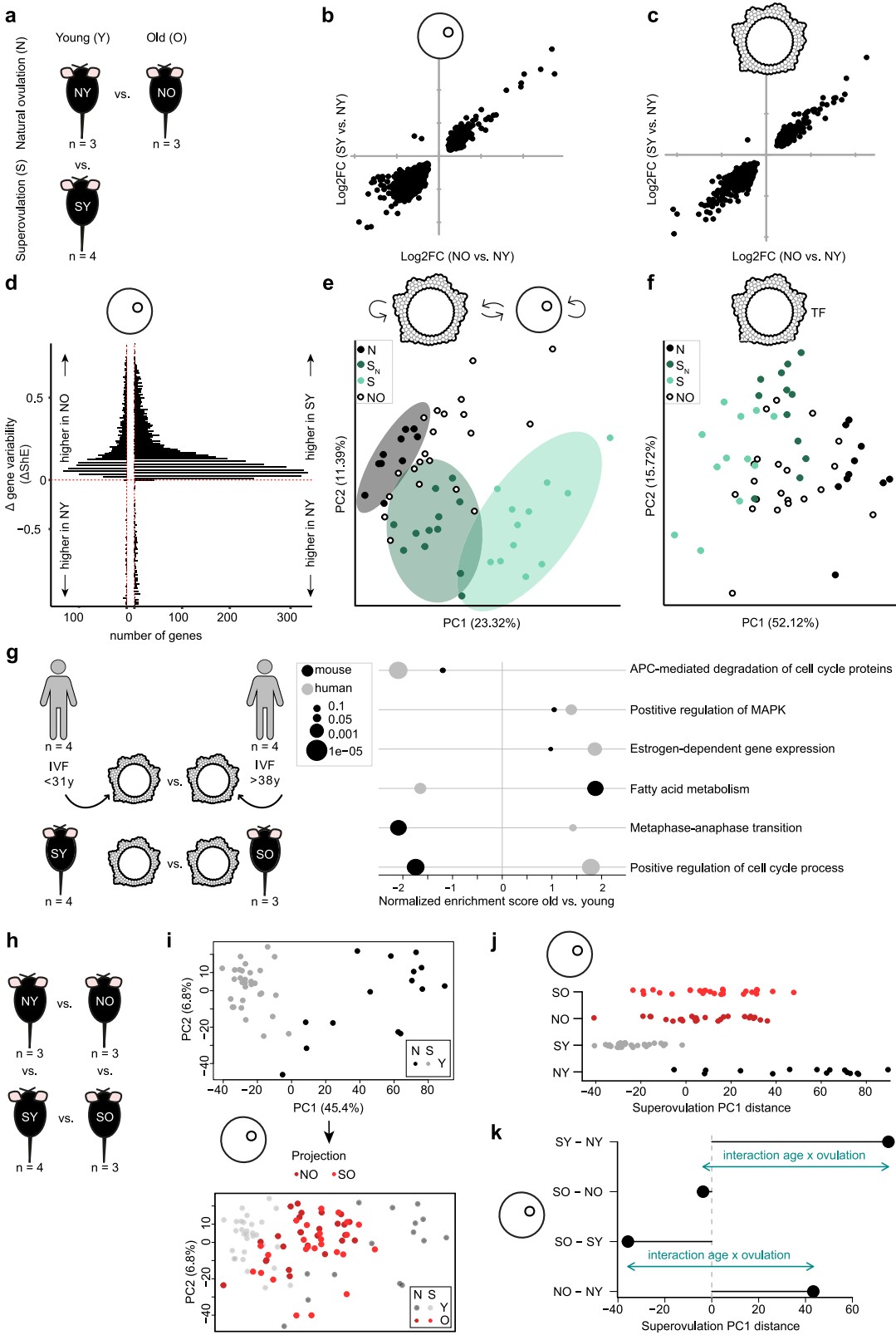

level could lead to meiosis defects. We show that maternal transcriptome remodeling is also impaired in superovulated oocytes. This could explain the delayed development observed in resulting mouse embryos[13,14,48] and cause mouse embryo developmental arrest by perturbing zygotic genome activation[61].

The widespread perturbations of oocyte-granulosa ligand-receptor expression we observed could be responsible for the abnormal

maturation of superovulated oocytes previously reported[6-8]. In granulosa cells, superovulation perturbs well-studied pathways like cGMP signaling, essential for meiosis resumption triggered by the LH surge (reviewed in Sun et al.[62]). Superovulated granulosa cells may fail to properly respond to the LH-mimicking hCG injection, which would impair the transmission of maturation cues to the oocyte. The efficiency of fertility treatments is limited by a sizeable fraction of

**Fig. 5 | Superovulation and aging lead to similar but non-additive transcriptional changes in cumulus-oocyte complexes. a** Schematic representation of natural aging and superovulation comparison (NY naturally ovulated young, SY superovulated young, NO naturally ovulated old). Log2 fold change of gene expression in SY versus NY compared to Log2 fold change in NO versus NY oocytes (**b**) and granulosa cells (**c**). Only genes that were significantly differentially expressed in both comparisons are shown (two-tailed Wald test implemented in DESeq2, adjusted $p < 0.05$, all genes are shown in Supplementary Fig. 4a ($n = 3$ NY, 4 SY, 3 NO mice). **d** Distribution of gene variability values calculated as differential Shannon entropy (ΔShE) in NO (left) and SY (right) oocytes in comparison to NY oocytes ($n = 15$ NY, 31 SY, 26 NO oocytes). Increased Shannon entropy is associated with increased gene expression variability. **e** PCA computed on ligand-receptor interaction scores between NY, SY, and NO cell pairs (clusters N, $S_N$, and S defined in Fig. 3). **f** PCA computed on the activity scores of the transcription factors shown in Fig. 3d in NY, SY, and NO granulosa cells (clusters N, $S_N$, and S defined in Fig. 3).

**g** Selected dysregulated pathways in granulosa cells during aging in mice and human patients: young−3 months for mice ($n = 4$), <31 years for human patients ($n = 4$), and old−12 months for mice ($n = 3$), >38 years for patients ($n = 4$). Full list of pathways is provided in Supplementary Data 8. Circle size represents adjusted $p$ values from an adaptive multi-level split Monte-Carlo scheme (fgsea package). **h** Schematic representation of the 4-way comparison of aging and superovulation (SO superovulated old). **i** PCA of young oocytes computed using differentially expressed genes between NY and SY mice (upper panel, $n = 15$ NY, and 31 SY oocytes), onto which old naturally and superovulated oocytes are projected (lower panel, $n = 26$ NO, and 26 SO oocytes). **j** PC1 coordinates of all four groups from PCA in (**i**). **k** Quantification of shifts in PC1 between groups contrasted in (**j**). For each group the mean value was used to compute the shifts. The interaction effect represents the non-additivity of aging and superovulation. Source data are provided as a Source Data file.

retrieved oocytes that are immature and often fail to be fertilized[63]. In vitro maturation of these oocytes is a promising solution to obtain more embryos without increasing the number of ovarian stimulation cycles (reviewed in De Vos et al.[64]), though the design of such protocols requires an understanding of when and where the oocyte-granulosa cell communication and oocyte maturation cues are lost.

It has been suggested that IVF efficiency could be improved by using granulosa cell transcriptomes to infer the quality of associated oocytes. However, prior attempts using human granulosa cells have struggled to identify robust gene expression signatures, in part because most such studies have been retrospective, used a diversity of stimulation methods and deployed different outcome metrics[24]. In contrast, our design is prospective, where granulosa cell transcriptomes are used to group embryos and then study their development. Moreover, we identified a subgroup of higher quality superovulated granulosa cells by including naturally ovulated COCs as a new reference point. Conversely, our granulosa cell classifier could successfully predict a population of embryos of apparently lower quality that included most arrested embryos and showed transcriptional developmental delay.

In ovaries of mice and humans, two waves of follicle development have previously been observed, and can be differentiated by their developmental appearance and anatomical localization[65,66]. First-wave follicles ovulate first, with an increasing frequency of second-wave follicles until mice exhaust the first-wave pool around three to 4 months of age. Since our young mice were 3 months of age, we speculated that the two groups of superovulated granulosa cells we observed might be linked to these two waves, yet the young naturally ovulated COCs did not subdivide. An interesting future direction would be to investigate how their transcriptomic signatures differ in adults and how they impact follicle quality.

Previous reports showing that aging is associated with lower oocyte quality have relied on comparisons among superovulated cells. By adding natural ovulation as a common reference, we newly reveal that aging and superovulation result in similar transcriptomic perturbations. For example, in aged oocytes, spindle assembly and mitophagy are perturbed[32,51]. Our results indicate that similar pathways are also impacted by superovulation. We show that maternal transcriptome remodeling is perturbed in both conditions, something that was also previously reported in aged mouse oocytes[31]. This convergence may be caused by increased gonadotropin levels, which increase during aging in both humans and mice[67–69]. Similarly, increasing the standard gonadotropin dose for superovulation negatively affects oocyte quality (reviewed in Bosch et al.[6]). Conversely, there is evidence that reducing the standard dosage of gonadotropins for IVF may improve embryo development rates. Thus, a milder gonadotropin stimulation could be developed to increase both the quality of superovulated COCs and the proportion of $S_N$ oocytes. Despite the similar trend observed in superovulation and aging, we

observed that superovulation has a greater transcriptional impact on younger COCs, potentially due to reduced sensitivity of old granulosa cells to gonadotropins[70]. Overall, any studies attempting to identify the impact of aging on oocyte quality should carefully consider the interaction effects of aging and superovulation; the effects of natural aging are best evaluated using naturally ovulated older female mice as a reference.

Our study has a number of limitations. First, our primary analysis was based on transcriptome-wide analyses. To mitigate this, we used two different protocols (SMART-Seq2 and total RNA sequencing) to address poly-adenylation bias and confirm the transcriptome remodeling results. We also confirmed gene expression differences between the two groups of superovulated granulosa cells using qPCR, a method that can be more easily used in future studies. In addition, we used an imaging-based protocol to confirm *Esr2* expression differences in natural versus superovulated cells. Second, naturally ovulated mice were housed with vasectomized male mice for a few nights, whereas superovulated mice were not. It appears unlikely that this alone would lead to such widespread differences in COC expression between the two conditions in young - but not in aged - individuals. Third, in our human analysis, granulosa cells were obtained by follicle puncture, and cannot be linked to a single oocyte.

In sum, our results demonstrate that aging and superovulation dysregulate similar pathways involved in fertility. It has long been hoped that IVF efficiency might be improved by exploiting granulosa cells to judge oocyte quality. Our study demonstrates that the transcriptional state of granulosa cells indeed reflects the quality of their associated oocyte accurately enough to achieve this goal in a well-controlled system.

## Methods
### Animals
All animal experiments were carried out according to governmental and institutional guidelines and approved internally by the animal welfare officer (Tierschutzbeauftragter, approval number DKFZ-366) and by the local authorities (Regierungspräsidium Karlsruhe, approval number G-238/19). Mice were kept at the DKFZ animal facility in Tecniplast GM500 IVC cages in groups of up to six mice under controlled light-dark cycle (12 h/12 h, from 7:00 to 19:00), at ambient temperature of 20−23 °C and 60−70% humidity. The animals had access to standard laboratory chow, water, and environmental enrichments ad libitum. None of the animals were involved in previous procedures. *Mus musculus* mice, specifically, CD-1 male and C57BL/6J female mice were purchased from Janvier, France, and allowed to adapt to animal facility conditions for at least 1 week, C57BL/6Ly5.1 female mice were bred in house at DKFZ animal facility. All female mice used in experiments for cell collection were euthanized by cervical dislocation, CD-1 male mice were euthanized by carbon dioxide when they were no longer suitable for sham-mating. Natural or superovulation was performed on young

(11–14 weeks) and old (50–58 weeks) female mice, where young mice were all C57BL/6J and old mice were all C57BL/6Ly5.1. Each experimental group contained 3–4 mice (Supplementary Data 1 and 7). Around half of old mice presented female reproductive tract pathologies and could not ovulate, therefore, they were not included in these calculations.

## Induction of ovulation

Ovulation by hormonal stimulation was induced by intraperitoneal injection of 5 international units (IU) pregnant mare serum gonadotropin (PMSG - Pregmagon®, IDT Biologika) and a second injection of 5 IU human chorionic gonadotropin (hCG - Ovogest®, MSD Animal Health) 48 h later[71]. The COCs were collected from the oviduct ampullas 14–16 h after hCG injection in young mice and 18–19 h in old mice. Superovulated mice were not housed with vasectomized males. SY mice yielded 15 healthy oocytes on average, whereas old mice yielded 11 (Supplementary Data 7). Oocytes were considered healthy if they had a spherical shape with a uniform translucent cytoplasm and were surrounded by expanded granulosa cells.

Natural ovulation was induced by mating C57BL/6 females with vasectomized CD-1 males[9]. The mice were caged in a 1:1 female-male ratio shortly before the beginning of the dark cycle. The next morning females presenting a copulatory plug were sacrificed for cell collection. Young mice were sacrificed 2–3 h after the beginning of the light cycle, and old mice after around 6h[72]. Females without plugs were examined each morning for three consecutive days. Males were allowed to rest for at least 1 day between successful sham-matings. Naturally ovulated young mice yielded 7.5 healthy oocytes on average, whereas old mice yielded 9 (Supplementary Data 7).

## COC extraction

Mice were sacrificed by cervical dislocation and the oviducts dissected (Supplementary Data 7). Under the stereomicroscope (SteREO Discovery.V12, Zeiss) ampullas were torn to release the COCs into a 96 µl M2 media (M7167, Sigma) drop under mineral oil (69794, Sigma) at room temperature. Then, 4 µl of pre-heated 500 µg/ml hyaluronidase (H3884, Sigma) diluted in M2 media (final concentration in the drop 20 µg/ml) was added to separate the COCs into single units. The COCs were incubated for 10–20 min at 37 °C and then mechanically separated into individual M2 drops using a 115–124 µm glass retransfer pipette (VRE-ID-TL-115-124-4, BioMedical Instruments). Individual COCs were then washed in M2 once and incubated for less than 5 min with enzymatic mix Accutase[38] (A6964, Sigma) at 37 °C to further separate the granulosa cells from the oocytes. Once separated, GCs (~50–200) were collected from these individual drops as a small bulk using a mouth pipette pre-filled with Dulbecco's phosphate-buffered saline (DPBS, 59321C, Sigma; Supplementary Movie 1). Oocytes were washed twice with M2 media and once with DPBS before collection. Cells were immediately flash frozen in liquid nitrogen in individual 0.2 ml thin wall PCR tubes (731-0679, VWR) and stored at −80 °C until further use (Supplementary Fig. 1a). Only oocytes with uniform translucent cytoplasm and spherical shape were collected. The media in the mouth pipette was changed between micro-manipulations of singularized COCs, granulosa cells and oocytes to ensure sample specificity. All samples were collected within 1.5 h after mouse dissection.

## SMART-seq2

Oocyte and granulosa cell samples were processed for full-length cytoplasmic RNA amplification using a slightly modified SMART-seq2 protocol[73]. Libraries for next generation sequencing were prepared using Nextera XT DNA library Prep Kit (FC-131-1096, Illumina). Briefly, cells were lysed with 0.2% Triton X-100 (93443, Sigma Aldrich) and reverse transcription was facilitated by Maxima-H Minus Reverse Transcriptase (EP0751, Thermo Fisher). Amplification of complementary deoxyribonucleic acid (cDNA) was achieved with 17 PCR cycles for oocytes and 20 cycles for granulosa cells. 1 ng of cDNA was used as template for library preparation, then tagmented, double indexed and amplified with 11 PCR cycles. All libraries were multiplexed into a single pool prior to sequencing. Quality control for sample preparation was performed using 4200 TapeStation High Sensitivity DNA and D1000 tapes (5067, Agilent), and Qubit 4 fluorometer High Sensitivity dsDNA assays (Q33231, Thermo Fisher). Samples that failed cDNA amplification were excluded. Samples were sequenced on NextSeq 550 using Single-Ended 75 bp High-Output kits (20024906, Illumina) at the DKFZ Open Sequencing Lab.

## Total RNA sequencing of oocytes

Oocytes from naturally and superovulated mice (Supplementary Data 7) were collected as described in the sections "Animals" and "Induction of ovulation". COCs were singularized as described in the "Methods" section "COC extraction". SMARTer Stranded Total RNA– Low Input Mammalian kit (634411, Takara) was used for lysis, cDNA amplification, and library preparation with 10 cycles for PCR-1 and 16 cycles for PCR-2. Samples were sequenced as recommended by the manufacturer on Novaseq 6000 (Illumina) with paired-end sequencing at the Institute of Clinical Molecular Biology (IKMB), Kiel, Germany.

## Targeted in vitro fertilization

Young C57BL6/J mice were naturally ovulated or superovulated and COCs (Supplementary Data 7) singularized as described above. Granulosa cells were immediately flash frozen and paired oocytes were divided into individual drops of CARD media (KYD-005-EX-X5, Cosmobiousa) under mineral oil (10029, Vitrolife) for fertilization. The oocytes were incubated at 37 °C, 5% $CO_2$ for 30–40 min prior to fertilization[74].

Frozen *Mus musculus* (C57BL6/J) male sperm was obtained from the DKFZ Transgenic Service. The sperm was frozen using Nakagata protocol[75] by dispersing spermatozoa from freshly dissected caudae epididymide, aspirating into straws pre-filled with Human Tubal Fluid (HTF, KYD-008-02-EX-X5, Cosmobiousa) media and freezing in liquid nitrogen. Sperm straws contained $6.41–10.6 \times 10^6$/ml progressive spermatozoids with $7.75–14 \times 10^6$/ml motility. The sperm straws were thawed and capacitated in Fertiup media (KYD-005-EX-X5, Cosmobiousa) for 30 min as recommended by the manufacturer.

Fertilization was achieved by combining 2.5 µl of activated sperm with 20 µl CARD media drops containing a single oocyte and incubated at 37 °C, 5% $CO_2$ for 3 h. Oocytes were then individually washed in HTF media drops and transferred to 20 µl HTF drops for overnight culture. After 24 h, the fertilization rate was evaluated. 2-cell stage embryos were then transferred to G-1 PLUS media (10128, Vitrolife) for further development. Early morula stage embryos were continuously collected based on an observation window of 62–64 h post-fertilization. For late blastocysts, fertilization media was additionally changed to G-2 Plus media (10132, Vitrolife) after 48 h post-fertilization and blastocysts collected at 108 h post-fertilization. Micrographs for each individual embryo (Supplementary Fig. 3a) were taken shortly prior collection using Discovery.V12 Stereoscope equipped with AxioCam MRc camera (Zeiss). Images were acquired at $3840 \times 2160$px resolution, processed using Zen Blue software (v.3.3., Zeiss), and cropped to center embryos using Microsoft Photos App. For collection, the embryos were washed in DPBS and flash frozen as in the method section "COC collection".

Granulosa cells and embryo samples were further processed as described in method section "SMART-seq2", except that cDNA amplification PCR cycle number for embryos was 16–17. The samples were sequenced on a Novaseq 6000 (Illumina) at the IKMB and on a NextSeq 2000 (Illumina) at the DKFZ Open Sequencing Lab at comparable sequencing depths.

## qPCR validation of classifier marker genes

We selected 12 target genes for qPCR validation from the S versus $S_N$ genes used by the classifier. qPCR primers were designed with the IDT PrimerQuest Tool (available online (https://eu.idtdna.com/PrimerQuest/Home/Index) using sequences from the NCBI nucleotide database, and were produced by Sigma-Aldrich (Supplementary Table 1). qPCR reactions ($n = 54$) were prepared using a PowerUp SYBR Master Mix (A25742, Applied Biosystems), 500 nM final concentration for forward and reverse primers each, and ~1 ng of cDNA. Samples ($n = 54$, from the targeted IVF experiment) were analyzed in duplicates using the manufacturer's Fast cycling mode on a QuantStudio 5 Real-Time PCR System equipped with a 384-well block (Thermo Fisher).

Data was processed using the Design & Analysis software 1 and 2 (Thermo Fisher) and R (v.4.0.0). Relative gene expression was analyzed using the ΔCt method with *Gapdh* as a reference.

## RNA processing from human granulosa cells

All patient procedures were approved by the local ethical committee of the University of Heidelberg, Germany (Ethikkommission der Medizinische Fakultät Heidelberg, study approval number S-602/2013). Patient procedures were carried out at the University Women's Hospital Heidelberg, Heidelberg, Germany. All patients were female (as defined by the presence of ovaries, uterus, and vagina), no information on gender was collected. Informed written consent about clinical procedures, pseudonymized data usage and completed clinical questionnaires were signed by all participating patients. Participants did not receive any compensations. Only samples from normal responders referred for IVF due to male or idiopathic subfertility were included (Supplementary Data 9).

Controlled ovarian stimulation was performed depending on the individual patient's situation using long GnRH agonist protocol or the GnRH antagonist protocol[76] (Supplementary Data 9). Granulosa cells were retrieved from the follicular fluid after transvaginal ultrasound-guided follicle puncture for IVF[76,77]. Shortly, the follicular fluid was transferred to 14-ml tubes (352001, Falcon), incubated at 37 °C, and then moved to a 100-mm culture dish on a heated table at 37 °C (L126 Dual Workstation, K-Systems). Mural granulosa cells were identified under a stereomicroscope (SMZ1500, Nikon), washed in Sydney IVF fertilization medium (K-SIFM-20, Cook Medical), and 2.5 µl of cells were transferred to tubes pre-filled with RNAlater (AM7020, Invitrogen) for storage at 4 °C. Granulosa cell samples were centrifuged at $5000 \times g$ for 5 min, the supernatant discarded and total RNA isolated using TRIzol (15596026, Life Technologies) according to the manufacturer's instructions. Total RNA was extracted from 6 younger (⩽31 years) and 6 older patient (⩾38 years). Further, RNA was cleaned with DNA-free DNA Removal Kit (AM1906, Invitrogen). ~1 µg of RNA was used to construct libraries with Truseq Stranded mRNA kit (20020595, Illumina) and barcoded with IDT for Illumina-TruSeq DNA and RNA UD Indexes (20040871, Illumina). Samples were sequenced on Nextseq 2000 (Illumina) with 100 bp paired-end sequencing at DKFZ Open Sequencing Lab.

## HCR RNA-FISH staining, microscopy, and image analysis

After natural ovulation or superovulation (Supplementary Data 7), COCs were collected and digested into individual units with hyaluronidase (as in "Methods" section "COC extraction"), omitting digestion in Accutase. SuperFrost microscope slides (631-0108, VWR) with 2 stick-on well separators were pre-coated with Poly-L-Lysine solution (P8920, Sigma Aldrich). The individual COCs were placed in the wells and fixed in 4% ice cold Formaldehyde (28908, Thermo Fisher) solution for 15 min. HCR staining was prepared according to manufacturer's recommendations (Molecular Instruments). Briefly, the HCR probes for *Areg* (NM_009704.4), *Npr2* (NM_173788.4), and *Esr2* (NM_207707.1) were designed by Molecular Instruments on demand.

Probe solution was prepared by adding 0.4 µL (1 µM stock) of each hybridization probe set into 100 µL pre-heated HCR Probe Hybridization Buffer at 37 °C and the samples were incubated in probe solution overnight (~24 h) at 37 °C. One COC per slide was stained without probes and used as a negative control. 3 pmol hairpins of channels B1, B2, B3 were individually snap-cooled and then mixed with Amplification Buffer. After washes, the samples were incubated in this hairpin solution overnight (~24 h) at room temperature. Hoechst 33342 (H3570, Invitrogen) was incorporated in the penultimate wash step to stain the nuclei. Slides were mounted with ProLong Gold Antifade Mountant (P10144, Invitrogen). Images were taken on a LSM780 inverted confocal microscope (Zeiss) using Plan-Apochromat 20x/0.8 lens and acquisition software Zen Black (v. 2.3, Zeiss) at the DKFZ Light Microscopy Facility.

To quantify the mean fluorescence intensity, the multi-channel images (Z-stacks with 10 µm step size) were processed using ImageJ (v.1.53). The DAPI channel was segmented to identify granulosa cell nuclei (minimum area~30 pixels, Find Maxima tool), while deleting the oocyte signal using setSlice function. The mean intensity of green (*Npr2*), red (*Esr2*), and far-red (*Areg*) signals were measured in defined DAPI areas, while subtracting the background signal through the Rolling Ball Background Subtraction function. The min-max fluorescence intensity thresholds were set manually and varied between samples (minimum threshold 30–70; maximum threshold 210–255). For each COC, cells that were more than two standard deviations above the mean were excluded. For each mouse and channel, the normalized intensity was computed by subtracting the average intensity of the negative control from the other cells. An average intensity was then computed for each COC. To test for treatment effect, a linear mixed model was fitted using the lme4 package (v.1.1.27.1) in R (v.4.0.0) using treatment as a fixed effect and mouse as a random effect. The model fit was compared to a reduced model without treatment using the anova function to obtain a *p* value. The Chi-square statistic was 4.1109 with one degree of freedom.

## Statistics and reproducibility

The study was designed for pairwise comparison to assess the impact of superovulation and aging independently from one another, the four possible conditions were thus collected.

When assessing the effect of the conditions, the different samples (e.g., COCs) from the same mouse were considered as technical replicates, except in Fig. 5k and Supplemental Fig. 4m. Samples from different mice were considered biological replicates. When assessing variability within a mouse, samples were treated separately.

No statistical method was used to predetermine sample size, however, a minimum of 3 individuals per experimental group was used for statistical testing and reproducibility, similar to previous studies[9,31]. Each experiment could be reproduced on a different experimental day, sometimes separated over a few years, for both mice and human patients. The sex of all individuals was female, as assigned by the presence of reproductive organs (ovaries, uterus, vagina). No information on gender identity of the human participants was obtained.

The experiments were not specifically randomized. Mice were obtained from an external provider on different dates. Each experimental group contains mice from different purchases. This was not tracked during analysis as the strain is highly inbred. Human patients could not be randomized due to the nature of treatments performed.

The Investigators were not blinded to allocation during experiments and outcome assessment. Blinding during data collection was not possible due to the protocols used. Analysis was performed on all groups simultaneously using the same scripts, therefore, blinding was either irrelevant or not possible.

The data exclusion criteria were based on fixed quality control metrics evaluated after sequencing. Mice oocyte, embryo, and

granulosa cell samples were discarded: if the number of reads was below 5000, if the number of genes detected was below 1000, or if the percentage of mitochondrial reads was above 5% (Supplementary Data 1). Ten samples were re-sequenced due to metadata mis-assignment. Human granulosa cell samples were filtered based on mitochondrial reads percentage (below 18%, corresponding to 80% percentile) to remove low quality samples (4 samples in total). To disregard outliers in the HCR RNA FISH staining quantification, cells that were more than two standard deviations above the mean fluorescence intensity were excluded.

## Sequencing quality control

Sequencing raw alignment files were demultiplexed by the DKFZ Genomics and Proteomics Core Facility. The adapters were then trimmed and the sequencing reads were aligned to the mm10 genome (GRCm38) using STAR (v.2.7.0f). Gene-barcode count matrices were then analyzed using R (v.4.0.0) and Seurat[78] (v.4.0.3).

Raw count tables for all replicates were merged and normalized together using the SCTransform function from Seurat with default parameters. PCA was computed using top 3000 most variable genes and UMAP was computed on the first 30 PCs. Samples were kept if the number of reads was above 5000, the number of genes detected was above 1000, and the percentage of mitochondrial reads was below 5% (Supplementary Data 1).

## Differential gene expression (DGE) analysis

DGE analysis was performed using DESeq2[79] (v.1.28.1) and a pseudo-bulk approach. Briefly, SCT-corrected counts of each mouse were summed across all cells from the same cell-type and used as the input for DESeq2. The effects of age and superovulation were modeled with an interaction term between age and superovulation.

## Overrepresentation analysis (ORA)

ORA was performed using a hypergeometric test. Gene sets used were downloaded from the MsigDB (https://www.gsea-msigdb.org/gsea/msigdb/) pathway collection (Hallmarks, Kegg, Reactome and GO Biological Process). P values were adjusted using the Benjamini-Hochberg procedure. Filtering for relevant keywords was then performed before manually inspecting the results.

Keywords used for oocytes: meiosis, spindle, checkpoint, chromosome segregation, estrogen, insulin, PI3Akt, MAPK, gonadotropin, DNA repair, oxidative, apoptosis, steroid, progesterone, aneuploidy.

Keywords used for granulosa cells: proliferation, cell_cycle, mitosis, G2M, steroidogenesis, estrogen, steroid, extracellular, repair, apoptosis, androgen, progesterone, fatty, lutein, mapk, kit, erk, egf, epidermal growth.

## Cell-cell interaction (CCI) analysis

To assess the CCI between OC and GC within each follicle we computed an interaction score for each pair based on the CellChat[80] ligand-receptor database and gene expression. Gene counts were corrected using the SCnorm method (v.1.10.0) to take into account gene length. Only the top 100 most expressed ligand and receptor genes were used for analysis. Interaction scores were computed by multiplying the expression of the ligand gene in the sender cell and of the receptor gene in the receiver cell. In case of receptors with multiple subunits, only the least expressed subunit was considered. For each OC-GC pair, communication was assessed between OC and GC in both directions, as well as GC-to-GC and OC-to-OC (autocrine signaling). P values were adjusted using the Holm procedure.

## Transcription factor activity analysis

Single-Cell Regulatory Network Inference and Clustering (SCENIC v.1.2.4, R v.4.3.0) was used to perform single-cell regulatory network analysis in granulosa cells[81] using the SCT normalized gene expression

values. Gene co-expression networks were determined using GENIE3, enriched transcription factor motifs were scored using RcisTarget, and regulon activity scores were calculated using AUCell. To assess which regulons overlap with the pathways found to be significantly dysregulated by superovulation, we used a Jaccard index to quantify the overlap between a regulons' target genes and the genes in each pathway. A hypergeometric test was used to assess the enrichment. Hierarchical clustering of transcription factors was performed using Euclidean distances and complete clustering method. P values were adjusted using the Benjamini-Hochberg procedure.

## Pathway activity scoring

Scoring of pathway activity was performed using the AUCell package[81] (v.1.10.0, R v.4.3.0) by calculating an enrichment score of each gene set within the top 20% most expressed genes in each cell. The gene sets used in this analysis were the same as gene sets used in the ORA.

## Total RNA-seq analysis

Raw sequence files were demultiplexed by the DKFZ Genomics and Proteomics Core Facility. The sequencing reads were aligned to the mm10 genome using the Cogent NGS Analysis Pipeline (CogentAP, Takara Bio, v.1.5.1). Gene-barcode count matrices were then analyzed using R (v.4.0.0) and Seurat (v.4.0.3).

Samples passed a quality filter if the number of genes detected was above 5000 and the percentage of ribosomal and mitochondrial reads was below 18 and 2.5%, respectively. Cell-type-specific marker gene expression was checked to exclude granulosa cell contamination.

Raw counts for all replicates were merged and corrected using the SCTransform function from Seurat. Genes of interest targeted by maternal transcriptome remodeling were identified based on previous literature. For each sample and each gene we computed a log2 fold change using the average expression of the gene in the other ovulation group (natural ovulation was used as a reference for superovulated samples, superovulation as a reference for naturally ovulated ones).

For the transcriptome-wide analysis, we used a recently published dataset by Lee et al.[43]. Genes were deemed impacted by one of the three processes if they passed the following criteria for differences between the GV and MII groups: (1) a geometric mean tail length change of more than 20 nucleotides (with an adjusted p value under 0.001) and representing at least 70% of the maximum tail length, and (2) at least 50 reads in the stage with the highest expression (GV if degraded, MII if re-poly-adenylated). Additionally, for the de-adenylated groups, the natural log ratio of read counts between MII and GV had to be positive or below -2 for the "stable" and "degraded" groups respectively (based on Lee et al.[43] classification).

## Granulosa cell classifier

We trained two support vector machines to classify the new granulosa cells from the IVF experiment into $S_N$ and S groups. We used the consensus $S_N$ and S clusters of granulosa cells from the original experiment as training and test set. The package caret (v.6.0-94, R v.4.3.0) was used to perform the classification using support vector machines with a linear kernel.

For the first classifier, we used the regulons identified using SCENIC as detailed above. Only transcription factors that were determined to regulate the pathways perturbed in superovulation (see "Methods" section "Transcription factor activity analysis") were considered in the analysis. We scored their regulons on the SCnorm normalized counts of young superovulated granulosa cells using AUCell with the top 2000 genes. These scores were used as input training for the first classifier. To further refine the selection of transcription factors we calculated which factors showed significantly differential activity scores between the $S_N$ and S groups using a generalized linear mixed model with random intercept. Activity scores were used as the

dependent variable, while group labels were used as the independent variable, and sample label as random effect. The model was fit under a beta distribution using the glmmTMB package (v.1.1.7). P values were corrected for multiple testing using the Benjamini-Hochberg procedure. Transcription factors that showed significantly different activity (adjusted $p < 0.05$) between the two groups were used as features to train the classifier.

For the second classifier, we used DESeq2 to identify differentially expressed genes between the $S_N$ and S groups, while taking into account mouse of origin. Differentially expressed genes that passed the thresholding (baseMean above 50, adjusted $p$ value below 0.01 and absolute log2 fold change above 0.7) were used to compute a PCA. PC 1 clearly separated the $S_N$ and S groups and was thus used to further filter the genes (loading above 0.09, roughly top 50% genes). Genes that passed these thresholds were used to train the second support vector machine classifier. Prior to training, features that showed near-zero variance, that were correlated or showed linear dependency were removed. A genetic algorithm was used to perform the final feature selection. To tune the cost hyperparameter, we used adaptive resampling.

Superovulated granulosa cells from the IVF experiment were then classified using both the transcription factor and the differential gene expression trained classifiers. If the classification label of the IVF granulosa cells was different between the two classifiers then the cells were labeled as "not assigned" (NA).

### IVF copy number analysis
Embryos which developed from matching fertilized oocytes were split in two groups based on their granulosa cell classification. The InferCNV package (v.1.17.0, R v.4.3.0) was used to detect CNVs in the embryos. Embryos derived from natural ovulation were used as a control group.

### Embryonic pseudotime analysis
Sequencing data from Xue et al.[47] was processed as described above and raw count tables were merged with count tables of our naturally ovulated embryos. Counts were normalized using the Seurat SCTransform function as described above. Genes linked to early embryonic development were selected if they met the following three criteria: (i) the gene is in the top 3000 highly variable genes identified by Seurat, (ii) the Spearman correlation between the gene's expression and embryonic stage (2-cell to blastocyst) is above 0.85 (adjusted $p < 0.01$), (iii) the gene has an absolute log2 fold change between 8-cell and morula, and between morula and blastocyst above 1 and in the same direction as the general correlation. Slingshot[82] (v.1.6.1, R v.4.0.0) was used to calculate pseudotime trajectories for embryos. Principal component analysis was used to perform dimensionality reduction. Full covariance matrix was used instead of clusters, and the principal curves fitted using the Slingshot function.

### Differential Shannon's entropy
Differential Shannon Entropy (dShE) was used to assess the differences in transcriptional heterogeneity between young and old naturally ovulated oocytes as well as superovulated and naturally ovulated young oocytes. Differential ShE was calculated using the EntropyExplorer package[83] (v.1.1, R v.4.3.0). Multiple testing correction was performed using the Benjamini-Hochberg procedure.

### Human data analysis
Sequencing raw alignment files were demultiplexed by the DKFZ Genomics and Proteomics Core Facility. The sequencing reads were aligned by the DKFZ Omics IT and Data Management Core Facility (ODCF) to the 1KGRef_PhiX genome (hs37d5 genome from the 1000 genomes project with the PhiX-sequence as an additional contig) using STAR (v.2.5.3a) and the DKFZ ODCF RNA-seqWorkflow pipeline (v.1.3.0). Gene-barcode count matrices were then analyzed using R (v.4.0.0). DGE analysis was performed

using DESeq2 (v.1.28.1) and age group as a predictor (below 31 or above 38 years old).

Gene set enrichment analysis was performed using the fgsea package[84] (v.1.14.0) using the same pathways as before (Hallmarks, Reactome, and GO Biological Process, see "Overrepresentation analysis") and log2 fold change from DESeq2. The same analysis was performed on the mouse SO vs SY DGE results as a comparison. All pathways that are significant (adjusted $p < 0.05$) in at least one species are available in the Supplementary Data 8. The same overrepresentation analysis was performed after removing top differentially expressed genes between cumulus and mural granulosa cells[85] from the human dataset, with very little change in the results.

For the comparison of human and mouse oocytes, count data from Ntostis et al.[54] was downloaded from GEO. Counts from MII oocytes were used. Differential gene expression and gene set enrichment analysis were performed as described above for the granulosa cells.

### Reporting summary
Further information on research design is available in the Nature Portfolio Reporting Summary linked to this article.

## Data availability
The primary data generated in this study have been deposited in the ArrayExpress database under accession numbers: paired oocytes and granulosa cells SMART-seq2: E-MTAB-13479; paired embryos and granulosa cells SMART-seq2: E-MTAB-13480; total RNA-seq in oocytes: E-MTAB-13474; processed human granulosa cell RNA-seq: E-MTAB-13496.Pseudonymized patient background clinical information are available under restricted access due to data protection of the participants. The access in compliance with data protection, and for research purposes only, may be obtained by contacting Dr. Julia Rehnitz (Julia.Rehnitz@med.uni-heidelberg.de) and will be processed within 3 months. Data will be available for ten years once access has been granted. Data that allows conclusions to be drawn about individuals will not be shared. This data will not be shared to third parties. The raw human RNA sequencing data are available under restricted access through the EGA database under accession number EGAD50000001210. Access may be granted for research purposes only and in compliance with data protection regulations after signing a collaboration contract with Dr. Julia Rehnitz. The use of this data will be limited to research on oocyte and granulosa biology, and no genotyping will be allowed. Data access requests have to be submitted through the EGA portal.The processed datasets are publicly available from Lee et al.[43] Supplementary Data S1 [https://ars.els-cdn.com/content/image/1-s2.0-S221112472400038X-mmc2.xlsx], Xue et al.[47] from ArrayExpress with accession number E-GEOD-44183, and Ntostis et al.[54] from NCBI Gene Expression Omnibus repository with accession number GSE164371.The additional data generated in this study are provided in the Supplementary Information/Source Data files. Sequencing quality controls, pairing of oocytes, embryos with their granulosa cells, cell type specific marker genes, overrepresentation analyses for filtered genes, gene set enrichment analyses, information about mice and basic patient background information are available in Supplementary Data 1–9. Source data is available for all relevant figures presented in this study. Source data are provided with this paper.

## Code availability
Code used in this study is available in github repository[86] [https://github.com/goncalves-lab/Daugelaite-Lacour-Winkler-et-al/tree/v1.0.0].

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

## Acknowledgements

We thank Open Sequencing Lab (DKFZ), Light Microscopy Facility (DKFZ), Institute of Clinical Molecular Biology (IKMB), U. Bender

(Universitäts-Frauenklinik Heidelberg), I. C. R. Delgado (EMBL), animal caretakers from ATV-108, U. Kloz, L. Ziegler, F. van der Hoeven and R. Brecht (DKFZ) for technical support and assistance. We thank P. A. Ginno for offering analysis suggestions, S. Del Prete and J. Cleland for resources. This work was supported by NCT/Helmholtz core funding (B270 to D.T.O.and B210 to A.G.); European Research Council (788937 to D.T.O.); Helmholtz junior group leader post (to A.G.); and the Deutsche Forschungsgemeinschaft (FI 2558/1-1 to A.G. and RE 3647/1-2 to J.R.).

## Author contributions

Conceptualization, K.D., P.L., A.G., D.T.O.; Methodology, K.D., P.L., I.W.; Investigation, K.D., P.L., I.W., N.S., A.S., M-L.K., F.C.; Formal Analysis, P.L., I.W.; Resources, A.T., X.P.N., A.V., J.R.; Data Curation, K.D., P.L., I.W.; Writing—Original Draft, K.D., P.L., I.W., A.G., D.T.O.; Writing—Review & Editing, K.D., P.L., A.G., D.T.O.; Visualization, K.D., P.L., I.W.; Project Administration, K.D.; Funding Acquisition, A.G., D.T.O., J.R.; Supervision, A.G., D.T.O. All authors had the opportunity to edit the manuscript, and all authors approved the final manuscript.

## Funding

## Competing interests

The authors declare no competing interests.
