## [Peer Review File · Nature Communications]

REVIEWER COMMENTS

Reviewer #1 (Remarks to the Author):

Daugelaite et al. have conducted extensive transcriptomic studies on mouse oocytes and cumulus granulosa cells to examine the transcriptomic impacts of gonadotropin stimulation in comparison to transcriptomic changes observed during aging. The study posits that ovarian hyperstimulation may have effects akin to aging.

The authors utilized various bioinformatics tools to analyze oocytes and cumulus cells harvested from young and old mice, both under normal ovulation and superovulation conditions. Notably, they identified distinct types of granulosa cells from stimulated ovaries, an intriguing observation that, regrettably, is not explored in further detail.

The paper proposes the use of transcriptomic changes as a metric to assess oocyte quality. While the predominant measure of fertility treatment success is live birth, the authors focus on morula formation, which, though interesting, is somewhat limited in its practical application. Typically, embryos are cultured to the blastocyst stage (day 5) before transfer, so it is by definition known what oocyte becomes a morula/blastocyst. The manuscript would gain considerably in value if the authors could demonstrate that their machine learning algorithm effectively predicts the likelihood of an oocyte (or an embryo) leading to a viable birth.

Overall, the findings are intriguing and prompt numerous subsequent inquiries (as seen below). However, the suggested pathways lack validation and are not sufficiently contextualized within established biological knowledge. Consequently, the translational potential of the research is somewhat limited.

Below are my comprehensive comments, which I hope will contribute to the enhancement of the manuscript.

1. You mention that one of the main novelties is your method to separate oocytes and cumulus cells. In clinical settings, COCs are routinely denuded prior to ICSI. Could you clarify how your method differs from this standard practice, and why it might be superior?

2. Introduction. The introduction should be nuanced to more accurately reflect the realities of assisted reproduction. IVF was primarily developed to treat infertility, not specifically to combat aging. Infertility is typically defined as the inability to achieve pregnancy after 12 months of regular, unprotected intercourse, and in many countries only women under the age of 40 can access publicly funded IVF. Statistically, the largest demographic receiving assisted reproduction is women under 35 years of age (ESHRE statistics). Therefore, to state the IVF is used to treat old infertile women is not correct.

3. Introduction. In the introduction, it's also important to correctly portray the efficiency of IVF. Human reproduction is naturally inefficient, primarily due to a high rate of aneuploidies, and this doesn't mean that IVF further reduces fertility / oocyte quality. Please provide a clearer explanation and evidence regarding the claim that superovulation leads to lower egg quality, especially in humans. The current references are animal-based. Please consider also that infertility patients often have conditions that affect fertility and oocyte quality, like endometriosis or polycystic ovaries. Therefore, quality of oocytes from IVF treatments is not only dependent on the gonadotropin stimulation used.

4. Introduction. The term 'post-mortem extraction of oocytes from ovaries' needs clarification. If you are referring to human organ donors, please specify this. The paper you cited used fertility preservation tissue, which does not equal to post-mortem, so the current wording is misleading and should be revised accordingly.

5. For Figure 1, consider using different colors to represent the four experimental groups in the PCA for more detailed visualization. Additionally, it would be beneficial to include a heatmap based on the top 100 most variable genes to better illustrate the overall grouping of the samples in the full dataset. In section D, could you clearly mark which samples correspond to the young, old, superovulated, and normal groups?

6. In Figures 2A and 2B, the enriched pathways seem to be a mix of KEGG, GO, and gene sets. Consider standardizing the method for a more cohesive analysis and include key genes for clarity. Also, it would be interesting to specify whether these pathways are associated with upregulated or downregulated genes. For Figures 2C and 2D, presenting an unbiased view of all processes affected by disrupted polyA regulation, rather than just a few selected ones, would be more informative. Regarding natural mRNA degradation, that may affect polyA tail length, how do you ensure that such degrading transcripts, still captured by random priming, don't skew your results on? As a validity check, could you demonstrate that the two sequencing methods yield comparable results for other genes? Lastly, are the differences shown in Figures 2E and 2F statistically significant?

7. Regarding Figure 3, could you clarify how your RNA-seq data specifically led to the focus on cellular communication? Was this focus due to an enriched gene set, GO, or KEGG pathway in earlier analyses? The heatmap in its current form does not provide detailed information on the actual signaling pathways, which would be highly beneficial. Could you cluster the heatmap and showcase examples of altered signaling? Additionally, what is the connection between the altered cellular communication and the transcription factors listed in sections C and D? In section E, you mention for instance altered estrogen signaling, yet none of the estrogen receptors are present in D. Is there a reason for this? Also, if superovulation mirrors aging, do aging mice exhibit similar diversity in granulosa cell types?

8. In Figure 4, the cartoon representation of a morula resembles a metaphase II (MII) oocyte. Please consider revising this for accuracy. The data interpretation would be more straightforward if a naturally ovulated mouse was included as a control. Currently, it's challenging to determine what constitutes a 'normal' embryo pseudotime due to the lack of a control group (and the fact the pseudotime is not an established endpoint of oocyte/embryo quality). Another potential control could involve correlating embryo pseudotime with live birth rates, to define what an optimal pseudotime is, although this might be more complex to implement. The machine learning approach to identify oocyte developmental competence based on cumulus cell transcriptomes is intriguing, but its real value would be demonstrated by predicting live births. Is there a correlation between the machine learning results and embryo morphology described in the supplementary figure? For section B, it is unclear to me what the authors try to show with random sampling 6 arrested embryos.

9. In Figure 5, sections B and C lack clarity on which genes are depicted. Could you clarify if these genes are related to those mentioned in Figures 2A and B? Please provide additional details such as GO annotations for these genes. Also, are the genes that are commonly regulated by gonadotropins and aging also a part of the panel used to classify the embryos by machine learning in 4C? In section D, could you elaborate on what negative variation in gene expression means? Additionally, why is the focus only on oocytes for gene variation when earlier sections emphasized granulosa cells as being more responsive to aging and gonadotropins?

The comparison with human samples seems to involve a different cell type – granulosa cells from follicular fluid, as opposed to cumulus granulosa cells. This discrepancy raises questions about the relevance and value of this comparison. Further, it would be beneficial to have information about the infertility causes in the patients, the ovarian stimulation protocols used, and the outcomes of the IVF treatments.

The concept of oocyte pseudotime in sections I and J is not clearly defined. What does this mean, how does it relate to oocyte competency, or could it merely reflect variation in postovulatory ageing? Given the novelty of these transcriptomic quality analyses, I suggest validating them with the unbiased total RNA-seq data that you have from a separate batch of animals to strengthen the

findings in D, I and J. You could also test the oocyte pseudotime on published datasets on human oocytes, which are many, and could help to better establish the meaning of this endpoint. Please see for example these papers:

-Single-cell profiling of transcriptomic changes during in vitro maturation of human oocytes
Takeuchi et al 2022

-Melatonin promotes the development of immature oocytes from the COH cycle into healthy offspring by protecting mitochondrial function Zou et al 2020

-Single human oocyte transcriptome analysis reveals distinct maturation stage-dependent pathways impacted by age Llonch et al 2021

-Changes in the Mitochondria-Related Nuclear Gene Expression Profile during Human Oocyte Maturation by the IVM Technique Yang et al 2022

-Differing molecular response of young and advanced maternal age human oocytes to IVM Reyes et al 2017

-Single-Cell Quantitative Proteomic Analysis of Human Oocyte Maturation Revealed High Heterogeneity in In Vitro-Matured Oocytes Guo et al 2022

-The impact of maternal age on gene expression during the GV to MII transition in euploid human oocytes Ntostis et al 2022

10. Overall, the manuscript seems to contain an excessive number of references for the journal's format, yet references to literature on oocyte ageing and maturation are few. A more focused and streamlined reference list would be beneficial. The figure legends require additional detail for clarity. Please expand these legends to include statistical information, the number of observations, and more explicit explanations of the figures' contents. In terms of visual presentation, the use of greyscale, particularly in PCA plots and other illustrations, decreases readability. Consider employing color to improve both visibility. The discussion section lacks depth and does not sufficiently contextualize the findings within established biological knowledge (for example the papers listed above). A notable omission is the discussion of mitochondria, which are often central to studies of oocyte quality and aging. In addition, two types of granulosa cells have been described in mice, and this should be discussed in the light of your observations. Addressing this and other relevant biological aspects would greatly enhance the manuscript's contribution to the field.

In summary, the potential interest and impact of this study would be significantly elevated by further exploration and practical application of the findings. Specifically, if the cumulus cell signatures could predict live birth outcomes, it would greatly enhance the clinical relevance of the research. Additionally, a more in-depth investigation into the two types of granulosa cells, particularly in the context of treatment and age (and reports on different granulosa cell types in mice), could provide valuable insights into ovarian biology. I also think that a dose-response study would be very interesting; would a milder gonadotropin stimulation results in COCs more akin to

natural conditions? These extensions of your current work could significantly broaden its appeal and utility within both the scientific community and clinical practice.

Reviewer #2 (Remarks to the Author):

This manuscript describes an analysis of the patterns of gene expression in oocytes and their associated follicular granulosa cells, comparing naturally ovulated to superovulated eggs and young to old. The key innovation is that each oocyte and its associated granulosa cells have been tracked as pairs, so the relationship between them can be examined. As well, eggs have been fertilized, enabling the development of the resulting embryo can be associated with the gene expression pattern of the granulosa cells that enclosed the egg. This is an ambitious and to my knowledge original approach that exploits state-of-the-art RNA-sequencing and bioinformatic tools. Several intriguing and potentially important results are presented, including:

Fig. 2A, B - Clear gene-expression differences between the granulosa cells (GC) and eggs of the SY vs NY mice.

Fig. 2C-F – interesting and well-explained approach to look at differences in polyadenylation of transcripts in the different groups.

Fig 4C – shows the potential predictive value with respect to embryo development of the GC pattern of gene expression, subject to the important caveat described below.

Fig. 5B, C – intriguing correlation between the disruptions of gene expression observed after superovulation and during aging.

Notwithstanding these clear strengths, there are several points that need to be clarified or potentially compromise the value of the findings.

Major:

It seems that the superovulation and natural collections may have been performed only once, with 3 or 4 animals in each cohort. This introduces some uncertainty into the results. Also, the females used for natural collections were housed with males, whereas those used for superovulation were

not. Although it may be improbable that housing with the male affected gene expression in the follicle, this design means that superovulation vs natural ovulation is not the only difference between the two groups of females.

Fig 4C shows that the development of embryos was correlated with the transcriptional landscape of the GCs surrounding the oocytes from which they were derived. This is a key result, as it could validate the hypothesis that these characteristics of the GCs can predict embryonic developmental potential. It needs to be clear, therefore, how the gene expression-based pseudotime position (X-axis) for each embryo was determined. If it corresponds to gene expression patterns known to be associated with specific stages of embryonic development, this is fine. But if it has been derived algorithmically (as seems to be the case), then it may have no relationship to actual embryo development. This would mean, unfortunately, that the hypothesis has not been validated.

Related to this, in Fig 5I the embryos derived from superovulated young (SY) oocytes are more advanced than those derived from natural young (NY) oocytes. This seems to contradict Fig 4C as well as the central hypothesis of the manuscript.

l. 120-125 describes genes whose expression is altered in the superovulated oocytes. As there is no transcription in the oocyte at this time, the differences in transcript levels between natural and superovulated oocytes could not be due to effects on gene expression. More importantly, the encoded proteins may not play a role during meiotic maturation – in particular, those involved in hormonal response, DNA damage and meiosis (depending on the meiotic stage they regulate). Hence, it is difficult to see the physiological consequences of altered transcript levels.

l. 155-160. The authors state that altered expression in the oocyte of genes involved in the assembly and function of the meiotic spindle may lead to abnormalities. This argument implies that the encoded proteins must be newly synthesized during maturation, as opposed to already existing in the oocyte prior to superovulation. This seems surprising to me and needs to be supported by references to the relevant literature. Otherwise, the same issue arises as above – the changes in transcript abundance may have no biological consequence.

The explanation of how the data shown in Fig 5D was obtained and plotted needs to be made clearer. How does it show that there is more variability in gene expression in old vs young oocytes?

Minor:

The data in Fig 4B showing that the 6/58 embryos that arrested before the morula stage had an associated abnormal or non-classifiable pattern of GC gene expression is intriguing and promising.

Nonetheless, 6 is a rather small sample size and in only 4 of these could the GC gene expression pattern be classified as abnormal.

Fig. 3 – the ligand-receptor pair analysis is an interesting approach, but really is only informative if restricted to those pairs that play a role in oocyte or granulosa cell development.

l. 165. The statement that the altered transcript levels of *Areg*, *Esr2* and *Npr2* likely affected oocyte meiosis is not justified by this data. It would need to be shown that meiotic maturation failed to occur.

l. 300-305. What do the authors mean by ‘insufficient maturation of the oocytes ... through disruption of the cGMP signal transduction pathway’? The decrease in cGMP leads to the activation of cyclin-dependent kinase-1, which initiates maturation. What do they believe is disrupted and with what consequence.

l. 305. ‘...transmit ovulation cues ...’ Do the authors mean maturation cues? The oocyte does not control ovulation.

Reviewer #1 (Remarks to the Author):

General

Daugelaite et al. have conducted extensive transcriptomic studies on mouse oocytes and cumulus granulosa cells to examine the transcriptomic impacts of gonadotropin stimulation in comparison to transcriptomic changes observed during aging. The study posits that ovarian hyperstimulation may have effects akin to aging.

The authors utilized various bioinformatics tools to analyze oocytes and cumulus cells harvested from young and old mice, both under normal ovulation and superovulation conditions. Notably, they identified distinct types of granulosa cells from stimulated ovaries, an intriguing observation that, regrettably, is not explored in further detail.

The paper proposes the use of transcriptomic changes as a metric to assess oocyte quality. While the predominant measure of fertility treatment success is live birth, the authors focus on morula formation, which, though interesting, is somewhat limited in its practical application. Typically, embryos are cultured to the blastocyst stage (day 5) before transfer, so it is by definition known what oocyte becomes a morula/blastocyst. The manuscript would gain considerably in value if the authors could demonstrate that their machine learning algorithm effectively predicts the likelihood of an oocyte (or an embryo) leading to a viable birth.

Overall, the findings are intriguing and prompt numerous subsequent inquiries (as seen below). However, the suggested pathways lack validation and are not sufficiently contextualized within established biological knowledge. Consequently, the translational potential of the research is somewhat limited.

Below are my comprehensive comments, which I hope will contribute to the enhancement of the manuscript.

We thank the reviewer for their suggestions which greatly improved our manuscript.

1. You mention that one of the main novelties is your method to separate oocytes and cumulus cells. In clinical settings, COCs are routinely denuded prior to ICSI. Could you clarify how your method differs from this standard practice, and why it might be superior?

We agree that our COC experimental separation protocol is similar to the denuding of oocytes undergoing ICSI. The novelty of our work is the careful tracking of both components (granulosa layer and oocytes - and indeed the oocyte-derived embryos) of individual COCs. This enables a more rigorous analysis of cell-cell communication between oocytes and their surrounding granulosa cells. Furthermore, the addition of oocytes and embryos derived from natural ovulation in young and aged individuals was an additional strategic novelty, allowing us to decouple the effects of aging and superovulation.

We have now revised our manuscript to better elaborate how our methods relate to clinical practice (Introduction, lines 43-62, 74-85), and how the inclusion of natural ovulation newly enabled analysis of cell to cell communication in different conditions (Introduction, lines 94-109, and Discussion, lines 405-423).

2. Introduction. The introduction should be nuanced to more accurately reflect the realities of assisted reproduction. IVF was primarily developed to treat infertility, not specifically to combat aging. Infertility is typically defined as the inability to achieve pregnancy after 12 months of regular, unprotected intercourse, and in many countries only women under the age of 40 can access publicly funded IVF. Statistically, the largest demographic receiving assisted reproduction is women under 35 years of age (ESHRE statistics). Therefore, to state the IVF is used to treat old infertile women is not correct.

To more accurately reflect how assisted reproduction technologies were initially developed and have been applied clinically, we have heavily revised our Introduction. We now present sequentially: the inherent limitations of natural reproduction (lines 24-28), the development of IVF (lines 28-39), and the more recent usage of IVF across a broader demographic window (lines 67-72, see also point 3 below).

3. Introduction. In the introduction, it's also important to correctly portray the efficiency of IVF. Human reproduction is naturally inefficient, primarily due to a high rate of aneuploidies, and this doesn't mean that IVF further reduces fertility / oocyte quality. Please provide a clearer explanation and evidence regarding the claim that superovulation leads to lower egg quality, especially in humans. The current references are animal-based. Please consider also that infertility patients often have conditions that affect fertility and oocyte quality, like endometriosis or polycystic ovaries. Therefore, quality of oocytes from IVF treatments is not only dependent on the gonadotropin stimulation used.

We have now revised our introduction to more accurately portray the numerous factors that shape IVF efficiency in human (lines 26-28, 31-38), while also listing the model organism evidence that superovulation may produce lower quality oocytes (lines 37-43). The new introduction text now reads:

"In humans, reproduction is naturally inefficient with high rates of aneuploidy in oocytes and frequent loss of early pregnancy^{1,2}. Common pathologies such as ovulatory disorders, endometriosis, occlusions of the fallopian tubes, uterine fibroids, or sperm anomalies further decrease fertility³. To treat infertility, assisted reproductive technologies such as ovarian stimulation (or superovulation) followed by in vitro fertilization (IVF) have been developed^{4,5}.

Human IVF typically involves ovarian stimulation using hormonal treatments, usually with gonadotropins, to induce the development of multiple oocytes. While this increases the number of oocytes available for retrieval, there is some concern that the rapid stimulation and altered hormonal environment might affect the quality of some oocytes^{6,7}. However, disentangling the effect of numerous individual factors (e.g. patient age, lifestyle, medical history, genetic background, and/or sperm quality) from the process of superovulation itself is extremely challenging in humans. Model organisms like rodents or livestock can be used as a better controlled system to study how oocytes respond to superovulation. In these organisms, studies based on morphological and biochemical characterization have confirmed that, when compared to natural ovulation, superovulation leads to the abnormal release of more immature oocytes^{8,9}, impaired oocyte epigenetics^{10,11}, and lower fertilization and implantation rates¹²⁻¹⁴."

4. Introduction. The term 'post-mortem extraction of oocytes from ovaries' needs clarification. If you are referring to human organ donors, please specify this. The paper you cited used fertility preservation tissue, which does not equal to post-mortem, so the current wording is misleading and should be revised accordingly.

We agree that this phrase was misleading, and it has been deleted from the revised Introduction section.

5. For Figure 1, consider using different colors to represent the four experimental groups in the PCA for more detailed visualization. Additionally, it would be beneficial to include a heatmap based on the top 100 most variable genes to better illustrate the overall grouping of the samples in the full dataset. In section D, could you clearly mark which samples correspond to the young, old, superovulated, and normal groups?

1. In Figure 1c, we have replaced the PCA plot with a UMAP to represent the four experimental groups and used different colors to facilitate visualization.
2. We have generated a heatmap based on Top 100 most variable genes for each cell type as recommended (Supplementary Figure S1b and S1c).
3. Initially, Figure 1d included only data from natural ovulation in young mice. We have now replaced this panel with a heatmap where all experimental groups are included and labeled.

6. In Figures 2A and 2B, the enriched pathways seem to be a mix of KEGG, GO, and gene sets. Consider standardizing the method for a more cohesive analysis and include key genes for clarity. Also, it would be interesting to specify whether these pathways are associated with upregulated or downregulated genes.

We have used KEGG, GO and Reactome databases because they contain different pathways which provide a more comprehensive overview when combined. Based on the reviewer's recommendation, we now include detail in the Methods section 'Overrepresentation analysis (ORA)' about how the pathways have been selected for presentation. Additionally, we have modified the Figures 2a and 2b to add information about up- or downregulated genes. We have also re-run the analysis independently for up- and downregulated genes which are shown in a new supplementary table (Supplementary Table 4, Tabs 1-4).

For Figures 2C and 2D, presenting an unbiased view of all processes affected by disrupted polyA regulation, rather than just a few selected ones, would be more informative. Regarding natural mRNA degradation, that may affect polyA tail length, how do you ensure that such degrading transcripts, still captured by random priming, don't skew your results on?

Since our first submission, an important transcriptome-wide study has been published that quantified polyA tail length and transcript level during oocyte maturation¹⁵. We have now re-analysed our data referencing the results of this study to identify genes that are re-polyadenylated, de-adenylated, or degraded. We find that these processes are widely perturbed in superovulated oocytes, and we added top genes from this reference to the main figure panel (Figure 2d); the full heatmap is available in Supplementary Figure S1d.

As a validity check, could you demonstrate that the two sequencing methods yield comparable results for other genes?

We thank the reviewer for the suggestion, we have now added such controls which we provide in the Supplementary Figure S1c and S1d.

Lastly, are the differences shown in Figures 2E and 2F statistically significant?

According to the suggestion, we have tested statistical significance among the groups using a Wilcoxon test. The groups are all statistically significant. We have now revised the Figures 2e and 2f to include the statistical significance.

7. Regarding Figure 3, could you clarify how your RNA-seq data specifically led to the focus on cellular communication? Was this focus due to an enriched gene set, GO, or KEGG pathway in earlier analyses?

The manuscript's topical transition was too abrupt at this point, which we have now corrected. A number of pioneering studies had focused on the importance of the cell-cell communication network for successful oocyte maturation¹⁶⁻¹⁸. The analyses in Figure 3 onwards were performed because we specifically designed our experiments to investigate at a transcriptome-wide scale the intercellular communication occurring between oocytes and granulosa cells. We have revised the text (Introduction, line 74 and Results, lines 199-202) to improve the story flow.

The heatmap in its current form does not provide detailed information on the actual signaling pathways, which would be highly beneficial. Could you cluster the heatmap and showcase examples of altered signaling?

To better showcase altered pathways in Figure 3a, we have now included examples of prominent signaling pathways that differ sharply between the classes of COCs. The rows have now been clustered based on the direction of the communication.

Additionally, what is the connection between the altered cellular communication and the transcription factors listed in sections C and D?

Figure 3 panels ab, cd, and e all demonstrate - using independent analytical approaches and gene sets - the existence of two COCs clusters following superovulation. For example, using only the top 100 ligands and receptors, panel 3a identified these two clusters, one of which more closely resembles naturally ovulated COCs. Similarly, an independent transcriptome-wide analysis, focused on the genes regulated by specific transcription factors, revealed the same subclusters in panel 3d.

In section E, you mention for instance altered estrogen signalling, yet none of the estrogen receptors are present in D. Is there a reason for this?

The analysis in Figure 3d identifies regulons with at least 20 direct target genes; since *Esr1* and *Esr2* did not pass this threshold, they were not included. Instead, in panel 3e we chose to showcase their function among the three clusters by scoring the activity of the entire estrogen signaling pathway, which includes both direct and indirect target genes. We have now reworded the main text to reflect this progression (Results, lines 214-216).

Also, if superovulation mirrors aging, do aging mice exhibit similar diversity in granulosa cell types?

It is true that both aged and superovulated granulosa cells show similar transcriptome perturbations compared to naturally ovulated young cells. However, the older granulosa cells do not split into two groups in cell-cell communication and transcription factor analyses.

8. In Figure 4, the cartoon representation of a morula resembles a metaphase II (MII) oocyte. Please consider revising this for accuracy.

We have replaced the cartoon with a more accurate representation.

The data interpretation would be more straightforward if a naturally ovulated mouse was included as a control. Currently, it's challenging to determine what constitutes a 'normal' embryo pseudotime due to the lack of a control group (and the fact the pseudotime is not an established endpoint of oocyte/embryo quality). Another potential control could involve correlating embryo pseudotime with live birth rates, to define what an optimal pseudotime is, although this might be more complex to implement. The machine learning approach to identify oocyte developmental competence based on cumulus cell transcriptomes is intriguing, but its real value would be demonstrated by predicting live births.

To address these excellent suggestions, we have now performed additional experiments and computational analyses.

First, experimentally, we have performed additional IVF experiments to include naturally ovulated embryos at morula and blastocyst stage and newly included blastocysts from superovulated mice. In total, these additional experiments have tripled the number of embryos used in Figure 4.

Second, computationally, to accurately understand where our embryos lie within the developmental trajectory, we now have replaced the pseudotime analysis with a new approach. Using a published dataset of early embryonic development (Xue et al.¹⁹) and our newly-added naturally obtained embryos, we built a reference gene expression time course and used it to order all our embryos more reliably (see revised results at page 7, lines 258-266, and Methods page 20, section "Embryonic pseudotime analysis"). These new experiments and computational approaches have created a much more intuitive developmental trajectory on which embryo development can be mapped.

The reviewer also suggested that we profile and quantify live births, in order to anchor the developmental trajectory. We agree that this would be very valuable and that it should be pursued in future studies. For this resubmission, we have not undertaken this experiment for two reasons. First, our power analysis indicates that the experimental design for such an experiment requires large numbers of IVF experiments using very large cohorts of donor and recipient mice (about three times larger than the total in the current study), in order to achieve statistical significance. The large number of replicates required is driven by the technical challenges this study would face in adequately controlling for the multiple variables involved in such IVF experiments. Second, the timeframe needed for such experiments greatly exceeds the resubmission deadline.

Is there a correlation between the machine learning results and embryo morphology described in the supplementary figure?

This is correct: poor embryo morphology correlates with the classification in Figure 4. Figure S3a shows the embryo morphology at collection 60-62h post-fertilization for morulas and 108h post-fertilization for blastocysts. The embryos marked in red have stopped developing, judging from their morphology. The embryos in the S or "not assignable" groups are slightly more likely

to arrest early on, however this was not statistically significant due to the relatively small number of embryos.

For section B, it is unclear to me what the authors try to show with random sampling 6 arrested embryos.

In the first submission, we showed what the distribution of class S_N , S and unclassified would be if the arrested embryo were not linked to the granulosa cells and thus randomly distributed among them.

In the resubmission, we have analyzed far more embryos. With these new replicates, we now compare the proportions of early arrested embryos between S and S_N to make this result more intuitive for the reader (Supplementary Figure S3c).

9. In Figure 5, sections B and C lack clarity on which genes are depicted. Could you clarify if these genes are related to those mentioned in Figures 2A and B? Please provide additional details such as GO annotations for these genes.

Yes, Figures 5b and 5c show how the transcription of same genes found in Figures 2a and 2B compare to aging. Sections 5b and 5c show only the genes that are dysregulated by both superovulation and aging; we have added the full set of genes in Supplementary Figures S4a and S4b.

Additionally, we now provide the up and down regulated pathway analyses involved in superovulation (Supplementary Table 4, Tabs 1-4) and aging (Supplementary Table 4, Tabs 5-8). We have done additional functional pathway analyses, including GO and KEGG, for genes depicted in Figure 5b and 5c (Supplementary Table 4, Tabs 9 and 10).

Also, are the genes that are commonly regulated by gonadotropins and aging also a part of the panel used to classify the embryos by machine learning in 4C?

We have further investigated the overlap between genes regulated by gonadotropins and aging versus the genes used for embryo classification in granulosa cells. We used two groups of genes for this classification: differentially expressed genes between the two groups (marker genes) and targets of selected transcription factors. We found that 18% of marker genes are impacted by aging and superovulation (12/65) while 6.6% of TF targets (646/9723) are. We now show this overlap in Supplementary Figure S4b.

In section D, could you elaborate on what negative variation in gene expression means?

Negative differential Shannon entropy means that transcriptional variability decreases relative to the reference; in panel 5d, this means a decrease in variability when compared to naturally ovulated young mice. We have rearranged the Figure 5d accordingly to facilitate its interpretation by adding 'higher in NY' and 'higher in NO/SY' to the Y-axes.

Additionally, why is the focus only on oocytes for gene variation when earlier sections emphasized granulosa cells as being more responsive to aging and gonadotropins?

We thank the reviewer for spotting this accidental omission. We performed the variability analysis for both oocytes and granulosa cells, and put the oocytes into the main text. However,

we forgot to add the corresponding granulosa cell analysis to the Supplemental section, which is now provided as Supplementary Figure S4e.

The comparison with human samples seems to involve a different cell type – granulosa cells from follicular fluid, as opposed to cumulus granulosa cells. This discrepancy raises questions about the relevance and value of this comparison.

In our dataset, human granulosa cells from follicular punctures are a mix of mural and cumulus granulosa cells. To ensure that our mouse-human comparison was relevant, we re-analysed the human dataset and focused the analysis purely on the gene sets that are similarly expressed in human cumulus and mural granulosa cells, based on work from Papler et al.²⁰ (Methods, page 21, lines 790-792). Because this analysis afforded highly similar results to what we reported in the original submission, we retained the original analysis approach while strengthening the text to indicate that the cell type discrepancy does not impact our results (Results section “Modest conservation of age-related changes in human samples”, lines 334-339).

Further, it would be beneficial to have information about the infertility causes in the patients, the ovarian stimulation protocols used, and the outcomes of the IVF treatments.

We now provide a Supplementary Table S9 with additional information about patient procedures, and amended the Method section accordingly (page 16, lines 584, 585). Only patients with male-caused or idiopathic subfertility were selected for these analyses to exclude possible confounding issues such as PCOS.

The concept of oocyte pseudotime in sections I and J is not clearly defined. What does this mean, how does it relate to oocyte competency, or could it merely reflect variation in postovulatory ageing?

The pseudotime plots in sections 5i and 5j were intended to show the non-additivity between aging and superovulation, but we see now that this presentation is not intuitive. Therefore, we have re-designed these panels to be more intuitive.

Our revised panel shows a PCA built on the young natural and superovulated samples using genes that are differentially expressed between the two groups, therefore capturing natural versus superovulation effects rather than postovulatory aging effects (Figure 5, panel i). We then projected the samples from aged mice onto this PC space and quantified the distance between all four conditions (Figure 5, panels j and k). This recapitulated the result presented in the original pseudotime analysis - specifically, the effects of aging and superovulation are non-additive. We have revised the text to reflect this new analysis (Results, lines 353-355, 361-363). The equivalent analysis performed on granulosa cells is available in Supplementary Figure 4k-m.

Given the novelty of these transcriptomic quality analyses, I suggest validating them with the unbiased total RNA-seq data that you have from a separate batch of animals to strengthen the findings in D, I and J.

According to the suggestion, we have now used total-RNA seq data, which are obtained from different animals, to confirm Shannon's entropy analysis in young oocytes from naturally and superovulated mice, the results are available in Supplementary Figure S4d.

You could also test the oocyte pseudotime on published datasets on human oocytes, which are many, and could help to better establish the meaning of this endpoint. Please see for example these papers:

- Single-cell profiling of transcriptomic changes during in vitro maturation of human oocytes Takeuchi et al 2022
- Melatonin promotes the development of immature oocytes from the COH cycle into healthy offspring by protecting mitochondrial function Zou et al 2020
- Single human oocyte transcriptome analysis reveals distinct maturation stage-dependent pathways impacted by age Llonch et al 2021
- Changes in the Mitochondria-Related Nuclear Gene Expression Profile during Human Oocyte Maturation by the IVM Technique Yang et al 2022
- Differing molecular response of young and advanced maternal age human oocytes to IVM Reyes et al 2017
- Single-Cell Quantitative Proteomic Analysis of Human Oocyte Maturation Revealed High Heterogeneity in In Vitro-Matured Oocytes Guo et al 2022
- The impact of maternal age on gene expression during the GV to MII transition in euploid human oocytes Ntostis et al 2022

We thank the reviewer for the suggestion. Because we have removed the Figure 5 pseudotime analysis, we have taken a different analysis approach to show how human oocytes correspond with our results in mice. In a new Supplementary Figure S4f, we show that the transcriptional impact of oocyte aging is remarkably similar between mouse and human, using the same approach as in granulosa cells in prior Figure 5f. As suggested by the reviewer, for this analysis we used a human dataset from Ntostis et al.²¹ that sequenced MII oocytes that were not *in vitro* matured from two age groups (Results, lines 341-344 and Methods, page 21, lines 793-795).

10. Overall, the manuscript seems to contain an excessive number of references for the journal's format, yet references to literature on oocyte ageing and maturation are few. A more focused and streamlined reference list would be beneficial.

We have adjusted the reference number to the journal's format (70 references in the main text) and included more focused literature on oocyte aging and maturation; see revised Introduction and Discussion sections.

The figure legends require additional detail for clarity. Please expand these legends to include statistical information, the number of observations, and more explicit explanations of the figures' contents. In terms of visual presentation, the use of greyscale, particularly in PCA plots and other illustrations, decreases readability. Consider employing color to improve both visibility.

We have included more detailed descriptions, often including statistical information, in figure legends of all five main text Figures, as well as the Supplementary figures. In many panels, we have redesigned figures where the choice of color and/or greyscale was not clearly distinguishable, for instance, Figures 1c, 2b, 2d, 2e, 5e, 5f, 5i, 5j.

The discussion section lacks depth and does not sufficiently contextualize the findings within established biological knowledge (for example the papers listed above).

We have re-written the entire Introduction and Discussion to contextualize the findings better, as well as incorporating suggested references.

A notable omission is the discussion of mitochondria, which are often central to studies of oocyte quality and aging.

We have now analyzed how genes in the mitophagy pathway differ between natural ovulation versus superovulation or aging, now shown in Supplementary Figure S4c (in text – Results, page 8, lines 295-300). Additionally, we now address mitochondria function in the Discussion (line 428).

In addition, two types of granulosa cells have been described in mice, and this should be discussed in the light of your observations. Addressing this and other relevant biological aspects would greatly enhance the manuscript's contribution to the field. In summary, the potential interest and impact of this study would be significantly elevated by further exploration and practical application of the findings. Specifically, if the cumulus cell signatures could predict live birth outcomes, it would greatly enhance the clinical relevance of the research. Additionally, a more in-depth investigation into the two types of granulosa cells, particularly in the context of treatment and age (and reports on different granulosa cell types in mice), could provide valuable insights into ovarian biology.

We thank the reviewer for this important comment, and we have now added discussion (lines 415-423) of the two waves of granulosa cell differentiation during embryonic development²²⁻²⁴.

We sought to add a more in-depth analysis of whether the two waves of granulosa cell development may be reflected in our study. To date, a few marker genes including *Foxl2* and *Lgr5* have been reported to differ in expression between the early and later waves. After birth, *Foxl2* and most other marker genes transcriptionally equalize between both granulosa cell waves; however, *Lgr5*, which is higher in the second wave, can still be observed upregulated until P5²². First wave follicles are the first to be ovulated, with an increasing frequency of second wave follicles until mice reach three to four months of age, when the first wave pool is exhausted. Whether the two waves of development can be transcriptionally distinguished in granulosa cells of adult ovarian tissues has not been studied.

Nevertheless, to further investigate these two waves, we performed a contained analysis of *Lgr5* expression in granulosa cells. We reasoned that granulosa cells from naturally ovulated 12 month old mice should contain only second wave granulosa cells, and could serve as a reference for *Lgr5* expression (see Reviewer Figure 1, column NO). 60% of samples from naturally ovulated young mice had relatively lower expression of *Lgr5*, consistent with their containing first wave granulosa cells. In young superovulated mice, all lowly-expressing cells were in the S_N group, while the S group only contained highly-expressing cells, presumably derived only from second-wave granulosa cells.

We felt that the conclusions we could draw from this analysis were too weak to include in the resubmitted manuscript. Limitations included that (1) the fate of these waves has not been

studied in adult, (2) our analysis rested on assumptions around only one gene's expression (*Lgr5*), and (3) there were conflicting data points in our analysis (e.g. superovulated old seemed to contain first wave granulosa cells).

Reviewer Figure 1: Proportion of granulosa cells expressing high or low *Lgr5*

If the reviewer feels this new analysis should be included then we would be happy to include it and further expand our Discussion section.

I also think that a dose-response study would be very interesting; would a milder gonadotropin stimulation results in COCs more akin to natural conditions? These extensions of your current work could significantly broaden its appeal and utility within both the scientific community and clinical practice.

Indeed, such a study would be interesting. We have now included citations about dosage effects in humans and rodents in the revised Introduction (lines 51-62) and added this point to the Discussion (lines 431-437).

Reviewer #2 (Remarks to the Author):

General

This manuscript describes an analysis of the patterns of gene expression in oocytes and their associated follicular granulosa cells, comparing naturally ovulated to superovulated eggs and young to old. The key innovation is that each oocyte and its associated granulosa cells have been tracked as pairs, so the relationship between them can be examined. As well, eggs have been fertilized, enabling the development of the resulting embryo can be associated with the gene expression pattern of the granulosa cells that enclosed the egg. This is an ambitious and to my knowledge original approach that exploits state-of-the-art RNA-sequencing and bioinformatic tools. Several intriguing and potentially important results are presented, including:

Fig. 2A, B - Clear gene-expression differences between the granulosa cells (GC) and eggs of the SY vs NY mice.

Fig. 2C-F – interesting and well-explained approach to look at differences in polyadenylation of transcripts in the different groups.

Fig 4C – shows the potential predictive value with respect to embryo development of the GC pattern of gene expression, subject to the important caveat described below.

Fig. 5B, C – intriguing correlation between the disruptions of gene expression observed after superovulation and during aging.

Notwithstanding these clear strengths, there are several points that need to be clarified or potentially compromise the value of the findings.

We thank the reviewer for their suggestions which greatly helped us improve our manuscript.

Major:

It seems that the superovulation and natural collections may have been performed only once, with 3 or 4 animals in each cohort. This introduces some uncertainty into the results.

There are three distinct experimental datasets performed fully independently, at different times, and using different mouse cohorts. Furthermore, our resubmission has now added two novel experimental datasets, that greatly increase the number of biological replicates. To briefly review:

The first cohort was used in the analyses reported in Figures 1, 2ab, 3, and 5; these analyses were performed on data collected from 97 OCs and 94 GCs originating from 3-4 animals per experimental group (NY-NO-SY-SO). The COCs from these mice were harvested from 1-2 individuals at a time.

The second cohort for the total RNA experiments in Figure 2cd used a fully independent set of 59 OCs from 4 NY and 3 SY mice. To confirm *Ers2* signaling with HCR, we used 2 NY and 1 SY mice.

The third cohort in our original manuscript used 52 embryos from IVF experiments, obtained from an independent cohort of mice. In response to the reviewers' comments, we have now greatly increased the number of replicates in the IVF section of our study. For our

resubmission, we expanded the sample size for the IVF experiments in Figure 4, by adding further 102 embryos and 120 granulosa cell samples from 13 additional independent mice.

The resubmitted manuscript now contains data derived from 40 mice, distributed as follows: 18 young naturally ovulated mice, 16 young superovulated, 3 old naturally ovulated, and 3 old superovulated mice. To summarize all the above, for our resubmission we have now amended the Supplemental Tables S1 and S8 to clearly list the number of mice as well as number of samples used, in each experimental cohort.

Also, the females used for natural collections were housed with males, whereas those used for superovulation were not. Although it may be improbable that housing with the male affected gene expression in the follicle, this design means that superovulation vs natural ovulation is not the only difference between the two groups of females.

We agree with the reviewer that not housing superovulated mice with vasectomized males could in principle impact the results. We have now specifically mentioned this potential confounder in the Methods section “Induction of ovulation”, page 13, and Discussion section where we list the caveats and limitations of our study (page 12, lines 444-455).

Fig 4C shows that the development of embryos was correlated with the transcriptional landscape of the GCs surrounding the oocytes from which they were derived. This is a key result, as it could validate the hypothesis that these characteristics of the GCs can predict embryonic developmental potential. It needs to be clear, therefore, how the gene expression-based pseudotime position (X-axis) for each embryo was determined. If it corresponds to gene expression patterns known to be associated with specific stages of embryonic development, this is fine. But if it has been derived algorithmically (as seems to be the case), then it may have no relationship to actual embryo development. This would mean, unfortunately, that the hypothesis has not been validated.

We agree with the reviewer that in the original Figure 4c, our algorithmically-derived pseudotime analysis was suggestive, but not explicitly linked to development; the other reviewer had a related comment regarding pseudotime (see above). We have now used a published time-course of early embryonic development from 2-cell to morula¹⁹ and combined it with embryos from our natural ovulation oocytes to build a reference gene expression developmental time course. We then used this to order all of our embryos (see revised results at page 7, lines 258-266, and Methods page 20, section “Embryonic pseudotime analysis”). The new analysis demonstrates that the S group yields developmentally delayed embryos, consistent with the hypothesis that GCs can predict developmental potential of the associated oocyte.

Related to this, in Fig 5l the embryos derived from superovulated young (SY) oocytes are more advanced than those derived from natural young (NY) oocytes. This seems to contradict Fig 4C as well as the central hypothesis of the manuscript.

In our original submission, it was unclear how to interpret the direction of pseudotime. To make this analysis more intuitive, we have now taken a different approach, resulting in a revised Figure 5i. Our revised panel begins by generating a PCA of the young natural and young superovulated samples (Figure 5i, top panel). We then projected onto this PCA the old natural and old superovulated samples (Figure 5i, bottom panel). Comparing the first principle

component among all these samples shows that oocytes from older mice land between young normal and young superovulated (Figure 5j). We then quantified the distance between these groups (Figure 5k). This analysis recapitulated the result presented in the original pseudotime analysis - specifically, the effects of aging and superovulation are non-additive - but we believe that it is now much more intuitive. We have revised the text to reflect this new analysis (Results, lines 353-355, 361-363).

I. 120-125 describes genes whose expression is altered in the superovulated oocytes. As there is no transcription in the oocyte at this time, the differences in transcript levels between natural and superovulated oocytes could not be due to effects on gene expression. More importantly, the encoded proteins may not play a role during meiotic maturation – in particular, those involved in hormonal response, DNA damage and meiosis (depending on the meiotic stage they regulate). Hence, it is difficult to see the physiological consequences of altered transcript levels.

Indeed, by the time we collect oocytes, they are in MII, which is transcriptionally inactive. However, when the mouse receives superovulation hormones at the beginning of our experiment, follicles in earlier developmental stages and their oocytes may still transcriptionally respond^{25,26}. The exact timing of transcriptional arrest relative to hormonal cues during oocyte maturation remains an actively debated topic^{26,27}. To address this, we have revised the Introduction (lines 63-73) to accommodate background scientific knowledge for easier interpretation of analyses done in our study.

I. 155-160. The authors state that altered expression in the oocyte of genes involved in the assembly and function of the meiotic spindle may lead to abnormalities. This argument implies that the encoded proteins must be newly synthesized during maturation, as opposed to already existing in the oocyte prior to superovulation. This seems surprising to me and needs to be supported by references to the relevant literature. Otherwise, the same issue arises as above – the changes in transcript abundance may have no biological consequence.

Although transcription is silent between the GVBD and MII stages, translation of a number of maternal mRNAs is upregulated at the GVBD-prometaphase transition and at MII stage large scale translation takes place^{28,29}. The translation increases of SAC genes *Bub1b* and *Mad2l1* as well as numerous cell cycle genes; in contrast, *Mad11l* translation decreases²⁹⁻³¹. For example, changes in *Mad2* gene expression levels directly interfere with correct chromosome segregation^{30,32}. We have revised the Discussion (lines 382-388) to address this important point.

The explanation of how the data shown in Fig 5D was obtained and plotted needs to be made clearer. How does it show that there is more variability in gene expression in old vs young oocytes?

We thank the reviewer for pointing this out. We have now added a much clearer explanation within the main text of how we calculated the variability between sample conditions using Shannon's Entropy (ShE) (Results, page 8, lines 301-306). Figure 5d shows the distribution of the significant values across all genes. Because most genes have a positive differential ShE, it means that their variability is higher in old compared to young samples. In addition, we have better annotated Figure 5d to make it more graphically intuitive. Specifically, the Y-axis now indicates where variability is lower versus higher.

Minor:

The data in Fig 4B showing that the 6/58 embryos that arrested before the morula stage had an associated abnormal or non-classifiable pattern of GC gene expression is intriguing and promising. Nonetheless, 6 is a rather small sample size and in only 4 of these could the GC gene expression pattern be classified as abnormal.

In our revised IVF experiment, we have greatly expanded total embryo numbers used in this analysis (Figure 4, Supplementary Table S1 and S8), therefore, the sample size of arrested embryos has also increased substantially (Supplementary Figure S3a), however, the results were not statistically significant.

Fig. 3 – the ligand-receptor pair analysis is an interesting approach, but really is only informative if restricted to those pairs that play a role in oocyte or granulosa cell development.

Based on this comment, as well as feedback from the other reviewer, we have reorganized Figure 3a, in order to showcase the ligand-receptor pairs that are most relevant to oocyte and granulosa development and function. We have now included examples of prominent signaling pathways that differ sharply between the classes of superovulated COCs.

I. 165. The statement that the altered transcript levels of *Areg*, *Esr2* and *Npr2* likely affected oocyte meiosis is not justified by this data. It would need to be shown that meiotic maturation failed to occur.

We agree that the statement was misleading, we have now changed it to (Results, page 5, lines 189, 190):

“In our data, superovulation impaired this hormonal response of granulosa cells, potentially delaying the transmission of the maturation cues (Figure 2f).”

I. 300-305. What do the authors mean by ‘insufficient maturation of the oocytes ... through disruption of the cGMP signal transduction pathway’? The decrease in cGMP leads to the activation of cyclin-dependent kinase-1, which initiates maturation. What do they believe is disrupted and with what consequence.

We observed that the expression of three key cGMP genes (*Areg*, *Npr2* and *Esr2*) was either higher or lower than expected in superovulated GCs when compared to naturally ovulated (Figure 2f). The specific disruption pattern is consistent with a delayed or impaired response of superovulation GCs to maturation cues³³. We have now re-written this section of the Discussion to more explicitly lay out how this pattern supports our hypothesis (lines 394-397).

In addition, using an orthogonal validation approach based on imaging (*in situ* hybridization), in independent experiments with a novel set of mice, we have now validated that *Esr2* expression is substantially increased following superovulation (Supplementary Figure S1f, Results, page 5, lines 190-194).

I. 305. ‘...transmit ovulation cues ...’ Do the authors mean maturation cues? The oocyte does not control ovulation.

Yes, this was a typo, and we have changed the sentence in the Discussion (lines 396, 397).

Reviewer references:

1. Martin, R. H., Ko, E. & Rademaker, A. Distribution of aneuploidy in human gametes: Comparison between human sperm and oocytes. *Am. J. Med. Genet.* **39**, (1991).
2. Wang, J. X., Norman, R. J. & Wilcox, A. J. Incidence of spontaneous abortion among pregnancies produced by assisted reproductive technology. *Hum. Reprod.* **19**, (2004).
3. Vander Borght, M. & Wyns, C. Fertility and infertility: Definition and epidemiology. *Clinical Biochemistry* vol. 62 (2018).
4. Steptoe, P. C. & Edwards, R. G. Birth after the reimplantation of a human embryo. *Lancet* vol. 2 (1978).
5. Niederberger, C. *et al.* Forty years of IVF. *Fertility and Sterility* vol. 110 (2018).
6. Bosch, E., Labarta, E., Kolibianakis, E., Rosen, M. & Meldrum, D. Regimen of ovarian stimulation affects oocyte and therefore embryo quality. *Fertility and Sterility* vol. 105 (2016).
7. Jamil, M. *et al.* Impact of the number of retrieved oocytes on IVF outcomes: Oocyte maturation, fertilization, embryo quality and implantation rate. *Zygote* **31**, (2023).
8. Blondin, P., Coenen, K., Guilbault, L. A. & Sirard, M. A. Superovulation can reduce the developmental competence of bovine embryos. *Theriogenology* **46**, (1996).
9. Lee, M. *et al.* Adverse effect of superovulation treatment on maturation, function and ultrastructural integrity of murine oocytes. *Mol. Cells* (2017) doi:10.14348/molcells.2017.0058.
10. Huffman, S. R., Pak, Y. & Rivera, R. M. Superovulation induces alterations in the epigenome of zygotes, and results in differences in gene expression at the blastocyst stage in mice. *Mol. Reprod. Dev.* **82**, (2015).
11. Marshall, K. L. & Rivera, R. M. The effects of superovulation and reproductive aging on the epigenome of the oocyte and embryo. *Molecular Reproduction and Development* (2018) doi:10.1002/mrd.22951.
12. Lee, S. T. *et al.* Influence of ovarian hyperstimulation and ovulation induction on the cytoskeletal dynamics and developmental competence of oocytes. *Mol. Reprod. Dev.* **73**, (2006).
13. Ertzeid, G. & Storeng, R. Adverse effects of gonadotrophin treatment on pre- and postimplantation development in mice. *J. Reprod. Fertil.* **96**, (1992).
14. Ertzeid, G. & Storeng, R. The impact of ovarian stimulation on implantation and fetal development in mice. *Hum. Reprod.* **16**, (2001).
15. Lee, K., Cho, K., Morey, R. & Cook-Andersen, H. An extended wave of global mRNA deadenylation sets up a switch in translation regulation across the mammalian oocyte-to-embryo transition. *Cell Rep.* **43**, (2024).
16. Fan, H.-Y. & Sun, Q.-Y. Chapter 12 - Oocyte Meiotic Maturation. in *The Ovary (Third Edition)* (eds. Leung, P. C. K. & Adashi, E. Y.) 181–203 (Academic Press, 2019). doi:https://doi.org/10.1016/B978-0-12-813209-8.00012-1.
17. Kidder, G. M. & Vanderhyden, B. C. Bidirectional communication between oocytes and follicle cells: Ensuring oocyte developmental competence. *Canadian Journal of Physiology and Pharmacology* (2010) doi:10.1139/Y10-009.
18. El-Hayek, S. & Clarke, H. J. Control of oocyte growth and development by intercellular

- communication within the follicular niche. *Results Probl. Cell Differ.* **58**, (2016).
19. Xue, Z. *et al.* Genetic programs in human and mouse early embryos revealed by single-cell RNA sequencing. *Nature* **500**, (2013).
 20. Papler, T. B., Bokal, E. V., Maver, A., Kopitar, A. N. & Lovrečić, L. Transcriptomic analysis and meta-analysis of human granulosa and cumulus cells. *PLoS One* **10**, (2015).
 21. Ntostis, P. *et al.* The impact of maternal age on gene expression during the GV to MII transition in euploid human oocytes. *Hum. Reprod.* **37**, (2022).
 22. Niu, W. & Spradling, A. C. Two distinct pathways of pregranulosa cell differentiation support follicle formation in the mouse ovary. *Proc. Natl. Acad. Sci. U. S. A.* **117**, (2020).
 23. Zheng, W., Zhang, H. & Liu, K. The two classes of primordial follicles in the mouse ovary: Their development, physiological functions and implications for future research. *Molecular Human Reproduction* vol. 20 (2014).
 24. Garcia-Alonso, L. *et al.* Single-cell roadmap of human gonadal development. *Nature* **607**, (2022).
 25. Cadenas, J. *et al.* Regulation of human oocyte maturation in vivo during the final maturation of follicles. *Hum. Reprod.* **38**, (2023).
 26. Morton, A. J., Candelaria, J. I., McDonnell, S. P., Zgodzay, D. P. & Denicol, A. C. Review: Roles of follicle-stimulating hormone in preantral folliculogenesis of domestic animals: what can we learn from model species and where do we go from here? *Animal* vol. 17 (2023).
 27. Bouniol-Baly, C. *et al.* Differential transcriptional activity associated with chromatin configuration in fully grown mouse germinal vesicle oocytes. *Biol. Reprod.* **60**, (1999).
 28. Chen, J. *et al.* Genome-wide analysis of translation reveals a critical role for deleted in azoospermia-like (*Dazl*) at the oocyte-to-zygote transition. *Genes Dev.* **25**, (2011).
 29. Luong, X. G., Daldello, E. M., Rajkovic, G., Yang, C. R. & Conti, M. Genome-wide analysis reveals a switch in the translational program upon oocyte meiotic resumption. *Nucleic Acids Res.* **48**, (2020).
 30. Homer, H. A. *et al.* Mad2 prevents aneuploidy and premature proteolysis of cyclin B and securin during meiosis I in mouse oocytes. *Genes Dev.* **19**, (2005).
 31. Wei, L. *et al.* BubR1 is a spindle assembly checkpoint protein regulating meiotic cell cycle progression of mouse oocyte. *Cell Cycle* **9**, (2010).
 32. Niaux, T. *et al.* Changing Mad2 levels affects chromosome segregation and spindle assembly checkpoint control in female mouse meiosis I. *PLoS One* **2**, (2007).
 33. Parrella, A. *et al.* High proportion of immature oocytes in a cohort reduces fertilization, embryo development, pregnancy and live birth rates following ICSI. *Reprod. Biomed. Online* **39**, (2019).

REVIEWER COMMENTS

Reviewer #2 (Remarks to the Author):

The authors have extensively revised the manuscript and added a considerable amount of new data that strengthens their arguments. In particular, the evidence of an association between embryo development and granulosa cell transcriptome is stronger. The manuscript remains rather dense, in large part because they tackle three subjects – (i) predicting embryonic development via granulosa cell transcriptome; (ii) impact of superovulation on embryo quality; (iii) effect of aging on embryo quality. They're well-integrated but the presence of the latter two dilute the impact of the first, which is the most interesting.

The main issue that remains is that the authors over-interpret the observed differences in mRNA quantity. They have measured mRNA quantity and inferred polyadenylation state under different conditions. But they have not tested whether there are any differences in the amounts of the encoded proteins or in cellular activity (eg, mitochondrial activity, DNA repair, timing of meiotic maturation). Numerous statements throughout the manuscript need to be toned down – examples below – to reflect this limitation of the experimental approach and data.

Specific points to address:

Do the authors really want to claim that superovulation produces poor-quality eggs, and to make this claim solely on the basis of mRNA analysis? This is essentially a call to stop all in vitro fertilization programs. Yet, for young women, IVF arguably is (almost as) efficient as natural conception, and the papers cited to support the argument of poor quality are more than 20 years old.

l. 30. There is no direct evidence that any intercellular communication pathways have been disrupted. Drawing this inference on the basis of mRNA quantities is not justified. See also title for Fig. 3.

l. 35. I thought the point was that superovulation and aging did not interact; at least were not additive. In any case, it is very difficult to understand mechanistically how superovulation and aging affect expression of the same genes.

l. 74. Maturation per se does not require communication, except for the transfer of cGMP. For example, cumulus cell-free oocytes will mature. Communication is required for oocyte growth, which precedes maturation, so maturation indirectly depends on communication.

Fig. 1d. Blue and red must have been inadvertently switched. Makes no sense that oocyte markers are down-regulated in oocytes.

l. 145. No evidence that the pathways have been functionally disrupted. Moreover, the suppl. Figure seems to show that every pathway is disrupted.

l. 182. No evidence that anaphase is delayed in superovulated eggs, and speculation on the basis of the available data is not justified.

l. 186-90. Maturation is triggered by degradation of existing cGMP, not by a decrease in its synthesis. This paragraph needs to be rewritten, and the present results interpreted, to match the current understanding of how maturation is initiated.

l. 195. This statement may literally be true, but does not capture the important point that the data do not show whether the pathways themselves have actually been affected.

Fig. 3a,b. I remain baffled why the authors report data regarding ligand-receptor expression for signaling pathways that have not been reported to play any role on oocyte or granulosa cell development. Do they believe that expression of the mRNAs is sufficient to conclude that the signaling pathway is operative and important?

l. 262. 'Embryos derived from natural ovulation were the most advanced'. If I understand the approach correctly, the embryos from natural ovulation were used to construct the time-line and therefore to define the most advanced condition. So the statement essentially is that 'the embryos defined as most advanced were the most advanced.' Needs rewriting or more explanation.

l. 378. 'perturbs maturation pathways, like hormonal response in granulosa cells, and meiosis and transcriptome remodeling in oocytes'. As these pathways were not directly examined, this statement is not justified.

l. 389-391. Considering that virtually all human IVF is carried out using hormonal priming – essentially superovulation, except that the eggs are collected before they are ovulated – this statement is an exaggeration.

l. 393. No evidence is presented that the superovulated eggs matured abnormally.

l. 398. The cited paper indicates that ~20% of oocytes collected for IVF are not fully mature. To describe this as ‘the large fraction’ is an exaggeration.

l. 430. Authors argue that increased gonadotropin levels in aged females may impact oocyte quality yet on l. 438 that older granulosa cells have reduced sensitivity to gonadotropins. These are apparently contradictory.

Reviewers #3 and #4 (Remarks to the Author):

The authors have made extensive adjustments and conducted additional experiments that have significantly improved the manuscript's flow and impact. This effort is greatly appreciated.

I have some minor comments that I believe would further enhance the clarity of the manuscript:

1. The validity check of the two sequencing methods, SMART-seq2 and total RNA-seq, requires further clarification.

My previous question might not have been clear enough. Could you demonstrate whether these two sequencing methods yield comparable results for other genes? Specifically, do SMART-seq2 and total RNA-seq identify similar differentially expressed genes, or does the choice of sequencing methods affect the genes identified? For example, would the genes presented in Figures 2a and 2b exhibit similar expression patterns if SMART-seq2 or total RNA-seq methods were used?

Additionally, please review the legends for Figure 2 and Supplementary Figure 1. Figure 2d refers to Supplementary Figure 1c, which does not correspond to the heatmap for SMART-seq2 and total RNA-seq. Also, it seems that the legends for Supplementary Figures 1d and 1e have been switched.

2. For clarity, it would be beneficial to specify in the results section that only granulosa cells from young superovulated COCs divide into two distinct clusters (clusters SN and S).

3. It would improve clarity if the genes presented in Figures 5b and 5c were highlighted in a different color on Supplementary Figure 4a.

Additionally, Figures 5b and 5c, as well as Supplementary Figure 4a, have different styles for their x and y-axis legends. Including the number of genes in the plots would enhance clarity. It would also be valuable to provide the number or proportion of significant genes, as well as for the DEGs, in the text. This numerical data would aid reader comprehension and interpretation of the findings.

More clarification is needed for Supplementary Figure 4b.

For Supplementary Figure 4c, please specify whether the heatmap is clustered by columns, rows or both. If the heatmap is not clustered by columns, it would be beneficial to show how the samples group when clustering is applied. Regarding lines 297-299, "When specifically looking at mitophagy pathway genes, we observed similar perturbations in response to superovulation and aging (Supplementary Figure S4c)", it is currently challenging to validate and support your conclusion, particularly if column clustering is not applied.

Figure 5i: The color legend in the lower panel of Figure 5i does not match the plot.

These minor revisions will further enhance the manuscript's clarity and overall quality.

Reviewer #2 (Remarks to the Author):

The authors have extensively revised the manuscript and added a considerable amount of new data that strengthens their arguments. In particular, the evidence of an association between embryo development and granulosa cell transcriptome is stronger.

The manuscript remains rather dense, in large part because they tackle three subjects – (i) predicting embryonic development via granulosa cell transcriptome; (ii) impact of superovulation on embryo quality; (iii) effect of aging on embryo quality. They're well-integrated but the presence of the latter two dilute the impact of the first, which is the most interesting.

The main issue that remains is that the authors over-interpret the observed differences in mRNA quantity. They have measured mRNA quantity and inferred polyadenylation state under different conditions. But they have not tested whether there are any differences in the amounts of the encoded proteins or in cellular activity (eg, mitochondrial activity, DNA repair, timing of meiotic maturation). Numerous statements throughout the manuscript need to be toned down – examples below – to reflect this limitation of the experimental approach and data.

We thank the reviewer for their help in improving our manuscript, we hope that they will find our further revisions, in particular highlighting the transcriptional nature of our study, adequate.

Specific points to address:

Do the authors really want to claim that superovulation produces poor-quality eggs, and to make this claim solely on the basis of mRNA analysis? This is essentially a call to stop all in vitro fertilization programs. Yet, for young women, IVF arguably is (almost as) efficient as natural conception, and the papers cited to support the argument of poor quality are more than 20 years old.

We thank the reviewer for calling our attention to this very important point. We have revised the Abstract (page 2, lines 26-29) to make it clear that our objective is to propose improvements to the efficiency of IVF and not in any way to call for a stop of IVF programs. We have also replaced the wording "IVF outcomes" with "IVF efficiency" throughout the text (page 11, line 413 and page 12, line 467).

With regards to the papers cited, we have now updated the reference list to include an additional review summarising the effects of superovulation on oocyte quality and further embryo development (Swain and Pool, 2008 in addition to Bosch et al., 2016). We have also added a citation to two more recent primary research papers in mice and humans: Lee et al. 2017 showed specific perturbations in the oocytes after superovulation in mice including the retrieval of less mature oocytes, perturbations in organelle ultrastructures and reduced mitochondria function. In patients, Magaton et al. 2023 carried out a comparison between natural IVF cycles and conventional stimulation cycles to show that the proportion of retrieved mature oocytes was higher after natural cycle IVF. These three citations are now in the Introduction, line 57.

I. 30. There is no direct evidence that any intercellular communication pathways have been disrupted. Drawing this inference on the basis of mRNA quantities is not justified. See also title for Fig. 3.

We have changed previous I30 to “We tested the hypothesis that superovulation disrupts oocyte maturation, revealing the key intercellular communication pathways dysregulated at the transcriptional level by forced hormonal stimulation.” (page 2, line 33). We have also changed the title of Figure 3 to “Superovulation dysregulates ligand-receptor expression and transcription factor activity in cumulus-oocyte complexes”.

The current state of the art to analyse intercellular communication infers it from the mRNA expression of ligand and receptor genes. We agree that there is a potential gap between mRNA levels and actual communication, however there is great value in this type of analysis as this method has been previously shown to accurately recapitulate actual ligand-receptor interactions. For example, Lee et al. 2024, *Science* has demonstrated that key communication pathways can be used to predict cancer patient response to immune checkpoint inhibitors. Wei et al. 2023, *Cell*, predicts ECM remodelling changes in mouse and monkey embryo cultures using cell-cell communication approach and then shows that this is functionality valid by generating a knock-out. Other field-driving studies are reviewed in Armingol et al. 2020, *Nature Reviews Genetics* “Deciphering cell–cell interactions and communication from gene expression”.

I. 35. I thought the point was that superovulation and aging did not interact; at least were not additive. In any case, it is very difficult to understand mechanistically how superovulation and aging affect expression of the same genes.

In the text we have used the terms ‘non-additivity’ and ‘interaction’ in the statistical sense. In statistics ‘non-additivity’ is a synonym for ‘interaction’ so indeed our results show that aging and superovulation interact. Interaction/non-additivity refers to a situation where the combined effect of these two factors is not equal to the sum of their individual effects. This indicates that the combined effect cannot be explained by simply adding the individual effects. We agree that our wording was confusing, we have now replaced ‘non-additivity’ in the Results title and section with the term “interact” (page 9, 355, 364; and page 12, 449).

Regarding the underlying cause of the similarity between aging and superovulation. Briefly, aging and superovulation result in similar transcriptomic perturbations. For example, our results indicate that aging and superovulation perturb the transcription of genes involved in spindle assembly, mitophagy, and maternal transcriptome remodeling. We propose that this similarity may be caused in both conditions by increased gonadotropin levels. This occurs in superovulation due to injection of exogenous gonadotropins, and in aging due to a natural increase previously described in both humans and mice. To explain the interaction effect we added to the Discussion (lines 445-447): “Despite the similar trend observed in superovulation and aging, we observed that superovulation has a greater transcriptional impact on younger COCs, potentially due to reduced sensitivity of old granulosa cells to gonadotropins”.

I. 74. Maturation per se does not require communication, except for the transfer of cGMP. For example, cumulus cell-free oocytes will mature. Communication is required for oocyte growth, which precedes maturation, so maturation indirectly depends on communication.

We have reworded this line to reflect this remark. It now reads: "Proper oocyte growth and maturation requires correct bi-directional communication with the surrounding cumulus granulosa cells." (current line 77)

Fig. 1d. Blue and red must have been inadvertently switched. Makes no sense that oocyte markers are down-regulated in oocytes.

Thank you for spotting this mistake, this has now been fixed!

I. 145. No evidence that the pathways have been functionally disrupted. Moreover, the suppl. Figure seems to show that every pathway is disrupted.

We have revised the paper to clearly indicate when disruption was observed on a transcriptional basis only. For example, current line 150 now reads: "Overall, superovulation transcriptionally disrupts pathways required for ...", also see lines 193 and 200.

We think the reviewer was referring to Table S4. This table reports only the set of pathways that are statistically significantly disrupted and does not report the unaffected classes. In total, we tested 9812 pathways, out of which 1311 were affected in oocytes and 1103 in granulosa cells. Most of these are not relevant for oocyte or granulosa function, which is why we added a keyword-based filtering step. We have now provided the full list of tested pathways and a column specifying which pathways were keyword-filtered.

I. 182. No evidence that anaphase is delayed in superovulated eggs, and speculation on the basis of the available data is not justified.

We have replaced the sentence "which together suggest a delayed anaphase onset in superovulated oocytes" with "which suggests differential SAC deployment in superovulated oocytes". (now in page 5, line 185-186).

I. 186-90. Maturation is triggered by degradation of existing cGMP, not by a decrease in its synthesis. This paragraph needs to be rewritten, and the present results interpreted, to match the current understanding of how maturation is initiated.

We have now revised the paragraph to make clear that the degradation of cGMP is in fact one mechanism for oocyte meiotic resumption (lines 189-193) and that we are only reporting transcriptomic deregulation.

I. 195. This statement may literally be true, but does not capture the important point that the data do not show whether the pathways themselves have actually been affected.

We have re-written this sentence from "In conclusion, superovulation perturbs germ and somatic cell transcription, affecting key COC maturation pathways" to "In conclusion, superovulation perturbs germ and somatic cell transcription of key genes involved in COC maturation pathways" (lines 199-200).

Fig. 3a,b. I remain baffled why the authors report data regarding ligand-receptor expression for signaling pathways that have not been reported to play any role on oocyte or granulosa cell development. Do they believe that expression of the mRNAs is sufficient to conclude that the signaling pathway is operative and important?

In this figure we chose to provide an unbiased presentation of our data without a priori filtering. We believe this analysis allows us to describe potentially new pathways with differential activity. This analysis does not only reflect the development, but the COC state at the time of ovulation, and this also includes broader processes, such as ECM remodelling and steroidogenesis. We are confident that this approach detected meaningful biological signals for the following reasons.

First, this analysis detects differential activity in many pathways known to play a role in oocyte or granulosa cell development. We will only comment on a few examples of outlined receptor-ligand pairs because it would be too exhaustive to comment on all of them. For example, in oocyte-to-oocyte communication Fgf1-Fgfr1/2 are known to activate the mitogen-activated protein kinase (MAP kinase) signalling cascade, partially induce extracellular signal-regulated protein kinase 2 (*Erk2*) phosphorylation and contribute towards meiosis re-initiation through Npr2 cascade (Cailliau et al., 2001). In oocyte-to-granulosa signalling, Bmp15-Bmpr1+2 are directly affected by gonadotropins and their mRNA levels should be lower in mature COCs (Guéripel et al., 2006), which was not the case for S group. In granulosa-to-oocyte signalling, Hspg2-Dag1 levels are important for oocyte quality and IVF outcomes. Lower perlecan (*Hspg2*) mRNA levels are associated with higher oocyte quality (Ma et al., 2020), as in N and, partially, S_N clusters. In granulosa-to-granulosa signalling, Wnt4-Fzd3+Lrp5/6, β -catenin canonical pathway is important for steroidogenesis in granulosa cells (Gifford, 2015).

Second, the current state of the art of analysing intercellular communication using gene expression of ligands and receptors has been previously shown to be highly reproducible at the protein level (see comment I.30).

I. 262. 'Embryos derived from natural ovulation were the most advanced'. If I understand the approach correctly, the embryos from natural ovulation were used to construct the time-line and therefore to define the most advanced condition. So the statement essentially is that 'the embryos defined as most advanced were the most advanced.' Needs rewriting or more explanation.

We thank the reviewer for bringing our attention to this section as the figure references were not adequately placed in the text. We have now fixed the text (page 7, lines 265-266) and figure legend. Briefly, our method is based on Figure 4c. Naturally ovulated morulas and blastocysts were used to extract a set of genes that show a strong and consistent increasing or decreasing level of expression over development (Figure 4c). For this analysis morulas were not defined as being more or less advanced, they were all taken together as a group and compared to blastocysts, which breaks circularity. Only then the pseudotime was generated by Slingshot (Figure 4d) to order our natural and superovulated embryos. We did not choose natural embryos as an end-point for this developmental trajectory.

I. 378. 'perturbs maturation pathways, like hormonal response in granulosa cells, and meiosis and transcriptome remodeling in oocytes'. As these pathways were not directly examined, this statement is not justified.

We have replaced "perturbs maturation pathways" with "perturbs gene expression in maturation pathways" (page 10, line 386).

I. 389-391. Considering that virtually all human IVF is carried out using hormonal priming – essentially superovulation, except that the eggs are collected before they are ovulated – this statement is an exaggeration.

We have now re-written this sentence to specify that these results and all references refer to mice: “This could explain the delayed development observed in resulting mouse embryos and cause mouse embryo developmental arrest by perturbing zygotic genome activation.” (page 11, lines 397-398).

I. 393. No evidence is presented that the superovulated eggs matured abnormally.

We have now reworded this sentence to: “The widespread perturbations of oocyte-granulosa ligand-receptor expression we observed could be responsible for the abnormal maturation of superovulated oocytes previously reported (Magaton et al., 2023; Swain and Pool, 2008; Bosch et al. 2016).” (line 401).

I. 398. The cited paper indicates that ~20% of oocytes collected for IVF are not fully mature. To describe this as ‘the large fraction’ is an exaggeration.

We softened the statement by replacing “large” to “sizeable” (line 406).

I. 430. Authors argue that increased gonadotropin levels in aged females may impact oocyte quality yet on I. 438 that older granulosa cells have reduced sensitivity to gonadotropins. These are apparently contradictory.

We have re-written the second sentence to clarify our argument (page 12, lines 445-450). First, we propose that a potential reason for the similarity between superovulation and aging is due to increased gonadotropin levels in aged females. And second, we propose that the reason that these two interact (superovulation has a greater transcriptional impact on younger COCs than in old) is potentially due to reduced sensitivity of old granulosa cells to gonadotropins.

Reviewer references:

1. Swain JE, Pool TB. ART failure: oocyte contributions to unsuccessful fertilization. *Hum Reprod Update*. 2008;14(5):431-446. doi:10.1093/humupd/dmn025
2. Bosch E, Labarta E, Kolibianakis E, Rosen M, Meldrum D. Regimen of ovarian stimulation affects oocyte and therefore embryo quality. *Fertil Steril*. 2016;105(3). doi:10.1016/j.fertnstert.2016.01.022
3. Lee M, Ahn J II, Lee AR, et al. Adverse effect of superovulation treatment on maturation, function and ultrastructural integrity of murine oocytes. *Mol Cells*. 2017. doi:10.14348/molcells.2017.0058
4. Magaton IM, Helmer A, Eisenhut M, Roumet M, Stute P, von Wolff M. Oocyte maturity, oocyte fertilization and cleavage-stage embryo morphology are better in natural compared with high-dose gonadotrophin stimulated IVF cycles. *Reprod Biomed Online*. 2023;46(4). doi:10.1016/j.rbmo.2022.11.008

5. Lee J, Kim D, Kong JH, et al. Cell-cell communication network-based interpretable machine learning predicts cancer patient response to immune checkpoint inhibitors. *Sci Adv.* 2024;10(5). doi:10.1126/sciadv.adj0785
6. Wei Y, Zhang E, Yu L, et al. Dissecting embryonic and extraembryonic lineage crosstalk with stem cell co-culture. *Cell.* 2023;186(26). doi:10.1016/j.cell.2023.11.008
7. Armingol E, Officer A, Harismendy O, Lewis NE. Deciphering cell–cell interactions and communication from gene expression. *Nat Rev Genet.* 2021;22(2). doi:10.1038/s41576-020-00292-x
8. Cailliau K, Browaeys-Poly E, Vilain JP. Fibroblast growth factors 1 and 2 differently activate MAP kinase in *Xenopus* oocytes expressing fibroblast growth factor receptors 1 and 4. *Biochim Biophys Acta - Mol Cell Res.* 2001;1538(2-3). doi:10.1016/S0167-4889(01)00074-X
9. Guéripel X, Brun V, Gougeon A. Oocyte bone morphogenetic protein 15, but not growth differentiation factor 9, is increased during gonadotropin-induced follicular development in the immature mouse and is associated with cumulus oophorus expansion. *Biol Reprod.* 2006;75(6). doi:10.1095/biolreprod.106.055574
10. Ma Y, Jin J, Tong X, et al. ADAMTS1 and HSPG2 mRNA levels in cumulus cells are related to human oocyte quality and controlled ovarian hyperstimulation outcomes. *J Assist Reprod Genet.* 2020;37(3). doi:10.1007/s10815-019-01659-8
11. Hernandez Gifford JA. The role of WNT signaling in adult ovarian folliculogenesis. *Reproduction.* 2015;150(4). doi:10.1530/REP-14-0685

Reviewers #3 and #4 (Remarks to the Author):

The authors have made extensive adjustments and conducted additional experiments that have significantly improved the manuscript's flow and impact. This effort is greatly appreciated.

I have some minor comments that I believe would further enhance the clarity of the manuscript:

We thank the reviewer for their help in improving our manuscript, we hope that they will find our further revisions adequate.

1. The validity check of the two sequencing methods, SMART-seq2 and total RNA-seq, requires further clarification.

My previous question might not have been clear enough. Could you demonstrate whether these two sequencing methods yield comparable results for other genes? Specifically, do SMART-seq2 and total RNA-seq identify similar differentially expressed genes, or does the choice of sequencing methods affect the genes identified? For example, would the genes presented in Figures 2a and 2b exhibit similar expression patterns if SMART-seq2 or total RNA-seq methods were used?

We only have total-RNA-seq for oocytes so we can only answer this question for the genes in Figure 2a. No, we do not expect the two technologies to detect the same genes as differentially expressed genome-wide. Oocytes at this stage have no new transcriptional activity, therefore, when using total-RNA we expect that the vast majority of genes will not show differential expression between superovulated and naturally ovulated oocytes. Biologically, the only instance in which we expect to see a difference in total-RNA is when transcripts are degraded (which according to literature seems to occur confidently in about 7% of the genes detectable with total RNA-seq). On the other hand, with SMART-seq2 (SS2) we expect to see differential expression in a larger set of genes including: genes whose transcripts are degraded, genes that are re-poly-adenylated and genes that are de-adenylated.

This is indeed the case. Despite having similar statistical power with both technologies (same number of samples assayed at a similar sequencing depth) and testing the same number of genes with the exact same statistical test, we find many more differentially expressed genes (DEG) with SS2 (51% of the tested genes at a false-discovery rate of 5%) than with total-RNA (5% of tested genes, Reviewer Figure 1). There is a reasonable degree of overlap between the two technologies (62% of DEG in total-RNA are also detected as DEG in SS2).

Reviewer Figure 1

We have compared the significantly differentially expressed genes detected with both methods. Using a false-discovery-rate-adjusted p-value threshold of 0.01, the results agree in fold-change direction for 80% of genes (Reviewer Figure 2).

Reviewer Figure 2

This agreement decreases with increasing adjusted p-value threshold as expected (as p-values grow the proportion of false positives in the DEGs increases, Reviewer Figure 3).

Agreement in fold change direction between Smart-seq2 and total RNA

Reviewer Figure 3

Among the genes detected as differentially downregulated in total RNA (5% of the genes tested), we observed a statistically significant enrichment in genes known to be degraded from previous studies, than compared to background (genes not targeted by re-polyadenylation, deadenylation, or degradation) (Reviewer Figure 4, Fisher-exact test p-value = $3.6e-6$).

Reviewer Figure 4

Note: the number of DEGs reported in Reviewer Figure 1 is larger than the one shown in Figure 2a because for the comparison with total-RNA we changed the normalisation method and statistical test so that we could use the exact same test for both SS2 and total-RNA (permutation test with the same normalisation approach). The overlap between the two tests on the same data is excellent (94% of genes in Figure 2a are also detected in the permutation test), but the permutation test is more sensitive and detects more genes as differentially expressed.

Additionally, please review the legends for Figure 2 and Supplementary Figure 1. Figure 2d refers to Supplementary Figure 1c, which does not correspond to the heatmap for SMART-seq2 and total RNAseq. Also, it seems that the legends for Supplementary Figures 1d and 1e have been switched.

We thank the reviewer for spotting this mistake! We have fixed the error in the legend in Figure 2d.

The Supplementary Figure 1d and 1e legends were not switched because Supplementary Figure 1d refers to a whole-transcriptome analysis from genes reported in Lee et al. (2024). The Figure 2d heatmap was subsampled from the heatmap in Supplementary 1e (known targets) for display purposes. We have re-written the Supplementary Figure 1e legend for more clarity.

2. For clarity, it would be beneficial to specify in the results section that only granulosa cells from young superovulated COCs divide into two distinct clusters (clusters SN and S).

We have now specified that this happens in young granulosa in the Results and Discussion sections (page 6, line 207, page 10, line 388).

3. It would improve clarity if the genes presented in Figures 5b and 5c were highlighted in a different color on Supplementary Figure 4a. Additionally, Figures 5b and 5c, as well as Supplementary Figure 4a, have different styles for their x and y-axis legends. Including the number of genes in the plots would enhance clarity. It would also be valuable to provide the number or proportion of significant genes, as well as for the DEGs, in the text. This numerical data would aid reader comprehension and interpretation of the findings.

Thank you for this suggestion, we have: added the number of tested genes, number of differentially expressed genes and proportion to the text (page 8, lines 291-295), we have highlighted the differentially expressed genes in Supplementary Figure 4a, and changed the axis legends.

More clarification is needed for Supplementary Figure 4b.

We have now rewritten the legend for Supplementary Figure 4b.

For Supplementary Figure 4c, please specify whether the heatmap is clustered by columns, rows or both. If the heatmap is not clustered by columns, it would be beneficial to show how the samples group when clustering is applied.

The heatmap was originally only clustered by row, and we have now also clustered it by column. We have replaced this figure and updated the figure legend accordingly.

Regarding lines 297-299, "When specifically looking at mitophagy pathway genes, we observed similar perturbations in response to superovulation and aging (Supplementary Figure S4c)", it is currently challenging to validate and support your conclusion, particularly if column clustering is not applied.

The analysis that we used in the results section is based on the pre-defined groups (differential expression between NY and SY, or NY and NO) that showed an enrichment for the mitophagy pathway. The heatmap allows the readers to better visualise how genes annotated as involved in mitophagy change between these conditions.

Figure 5i: The color legend in the lower panel of Figure 5i does not match the plot.

Thank you, we have amended the colour in the legend.

These minor revisions will further enhance the manuscript's clarity and overall quality.

REVIEWERS' COMMENTS

Reviewer #2 (Remarks to the Author):

The authors' clarification at various points in the ms that the results observed are at the level of transcription provide an adequate level of caution with respect to the interpretation of the results. The big-picture message - that GC transcriptome might predict oocyte competence - is very interesting and well-supported.

Reviewer #3 (Remarks to the Author):

The authors have now addressed all of my concerns.

Reviewer #4 (Remarks to the Author):

The authors have now addressed all of my concerns.